# Ego-centric Learning of Communicative World Models for Autonomous Driving

## Abstract

We study multi-agent reinforcement learning (MARL) for tasks in complex high-dimensional environments, such as autonomous driving. MARL is known to suffer from the *partial observability* and *non-stationarity* issues. To tackle these challenges, information sharing is often employed, which however faces major hurdles in practice, including overwhelming communication overhead and scalability concerns. Based on the key observation that world model encodes high-dimensional inputs to low-dimensional latent representation with a small memory footprint, we develop *CALL*, Communicative World Model, for ego-centric MARL, where 1) each agent first learns its world model that encodes its state and intention into low-dimensional latent representation which can be shared with other agents of interest via lightweight communication; and 2) each agent carries out ego-centric learning while exploiting lightweight information sharing to enrich her world model learning and improve prediction for better planning. We characterize the gain on the prediction accuracy from the information sharing and its impact on performance gap. Extensive experiments are carried out on the challenging local trajectory planning tasks in the CARLA platform to demonstrate the performance gains of using *CALL*.

## 1 Introduction

Many multi-agent decision-making applications, such as autonomous driving Kiran et al. (2021), robotics control Kober et al. (2013) and strategy video games Kaiser et al. (2019) require agents to interact in a high-dimensional environments. In this work, we study distributed reinforcement learning (RL) for autonomous driving Zhang et al. (2021a); Busoniu et al. (2008), where each agent carries out ego-centric learning with communication with other agents in the proximity Claus & Boutilier (1998); Matignon et al. (2012); Wei & Luke (2016). Departing from conventional stochastic game-theoretic approaches for studying equilibrium behavior in partially observable stochastic games (POSGs), which would not applicable to real-time applications de Witt et al. (2020); Matignon et al. (2012), we focus on the more practical setting that each agent is *ego-centric* and chooses actions to maximize her *own interest* Ozdaglar et al. (2021); Brown (1951). We propose Communicative World Model (*CALL*) to tackle two notorious challenges in multi-agent RL, namely *partial observability* and *non-stationarity*.

In multi-agent systems, information sharing has long been recognized as a crucial technique for improving decision-making Liu & Zhang (2023); Jiang & Lu (2018); Zhang et al. (2021a); Foerster et al. (2016); Sukhbaatar et al. (2016). For instance, agents can reduce uncertainty and improve coordination in complex environments by exchanging observations through a central server Oliehoek et al. (2008); Lowe et al. (2017); Wang & Meger (2023) or directly sharing with other agents Sukhbaatar et al. (2016); Jiang & Lu (2018); Foerster et al. (2016). However, in high-dimensional environments, simply sharing raw observations or state information does not scale efficiently Canese et al. (2021); Dutta et al. (2005). Therefore, more *efficient and targeted* information sharing strategies are critical to enabling scalable and effective multi-agent learning in high-dimensional environments.

Another significant challenge arises from the non-stationarity when other interacting agents adapt their policies in response to one another Moerland et al. (2023); Silver et al. (2017). This issue becomes even more pronounced in high-dimensional environments, where the interactions between agents become increasingly complex and grow exponentially in the number of agents. Clearly, direct

prediction of these intertwined environment dynamics in such high-dimensional spaces becomes computationally intractable Qu et al. (2020); Zhang et al. (2021a). A promising approach to address this challenge is to leverage the generalization of world models (WMs), which learn latent dynamics models in a much lower-dimensional latent space. However, recent works on WM based reinforcement learning still face significant limitations, and often rely on rigid, static information-sharing mechanisms, such as sharing information with all agents Pretorius et al. (2020), using centralized frameworks Krupnik et al. (2020); Pan et al. (2022); Liu et al. (2024), or adopting heuristic approaches that limit sharing to neighboring agents Egorov & Shpilman (2022). These systems will likely struggle with *scalability* and lack the *adaptability* needed for real-time decision making in the non-stationary environments. In order to harness the power of world models and address the challenges of partial observability and non-stationarity in high-dimensional environments, it is therefore of great interest to develop a decentralized, adaptive communication strategy that allow agents to share only the more relevant latent information. This paper seeks to answer the following question:

*How to synergize WM's generalization capability with lightweight information sharing for enhancing Ego-centric MARL in high-dimensional, non-stationary environments?*

To this end, we propose Communicative World Models (*CALL*) for ego-centric MARL, where each agent makes decisions while utilizing lightweight, prediction-accuracy-driven information sharing to enhance and strengthen its world model learning. The proposed *CALL* method is built on three key ideas: 1) *Latent state and intention representation.* In *CALL*, agents encode high-dimensional sensory inputs, such as camera images, into compact latent states that represent the key features of the environment. Agents also encode their planned actions as *latent intentions* that capture their future goals (i.e., the waypoints in planning tasks). These latent representations are more efficient to share and require only a fraction of the memory compared to the raw data. For instance, as summarized in Table 1, the

Table 1: Comparison of Bandwidth and Look-ahead Prediction Accuracy (%) Between Baseline Methods and *CALL*

|            | Raw Inputs | *CALL* |
|------------|------------|--------|
| Bandwidth↓ | 5MB        | 0.11MB |

| Pred. Accu.↑ | w/o sharing | *CALL* |
|--------------|-------------|--------|
| 5 steps      | 75%         | 87%    |
| 30 steps     | 63%         | 80%    |
| 60 steps     | 58%         | 72%    |

latent representations generated at each time step require only 1/50th of the bandwidth compared with a single high-definition sensor image, making lightweight information sharing much more feasible in practice. 2) *Prediction-accuracy-driven information sharing.* This low-overhead communication is further enhanced by prediction-accuracy-driven sharing, where agents judiciously exchange latent states and intentions based on their impact on improving prediction accuracy. This adaptability ensures that agents focus on sharing information that are more relevant to a better decision-making, avoiding the inefficiencies caused by transmitting unnecessary data. 3) *Synergization of WM's generalization capability with information sharing.* The generalization capability of world models, together with information sharing, ensures that agents can achieve high prediction accuracy while minimizing communication overhead. As illustrated in Table 1, the information sharing in *CALL* significantly improves prediction accuracy of the future latent state compared to the baseline without information sharing across by a notable margin.

Our main contributions can be summarized as follows:

- By synergizing world model's generalization with lightweight information sharing, we propose *CALL* to tackle two key challenges in MARL, namely partial observability and non-stationarity. Specifically, in *CALL*, each agent uses a world model to encode high dimensional data into low-dimensional latent representation, thereby facilitating information sharing among agents and improving learning the dynamics. The predictive capability, enabled by world models, allows agents to plan and make decisions that go beyond the current environment.

- To provide the guidance on information sharing in *CALL*, we characterize the prediction performance, and systematically study the impact of the generalization error in the world model and the epistemic error due to partial observability and non-stationarity. Guided by the theoretical results, we propose a prediction-accuracy-driven information sharing strategy which allows agents to selectively exchange the most relevant latent information. This adaptive sharing scheme is designed to reduce prediction error and minimize the sub-optimality gap in the value

Table 2: Related works in terms of (1) World Model, (2) State Sharing, (3) Intention sharing and (4) Information Sharing Mechanism.

| Paper | World Model | State Sharing | Intention Sharing | Information Sharing Mechanism |
|---|---|---|---|---|
| Das et al. (2019); Ma et al. (2024) | | √ | | $k$-hop neighbors |
| Jiang & Lu (2018); Kim et al. (2020) | | √ | √ | Attention Mechanism |
| Egorov & Shpilman (2022) | √ | √ | | Neighboring agents |
| Pretorius et al. (2020) | √ | √ | √ | All agents |
| **This Work** | √ | √ | √ | **Prediction Accuracy guided** |

function. Furthermore, *CALL* leverages WM's generalization capability to significantly improve the prediction of environment dynamics.

- To showcase the effectiveness of *CALL*, we conduct extensive experiments on the trajectory planning tasks in autonomous driving. The results shows that *CALL* can achieve superior performance with lightweight communication. Ablation studies further validate the importance of information sharing in addressing key MARL challenges, aligning with our theoretical predictions. Additionally, experiments in more complex environments highlight the *CALL*'s potential for scalability in real-world applications. To the best of our knowledge, our work is the first attempt to use world-model based MARL to solve autonomous driving tasks in the complex high-dimensional environments.

## 1.1 RELATED WORK

**World Model (WM).** WMs are emerging as a promising solution to model based learning in the high-dimensional environment Ha & Schmidhuber (2018); Hafner et al. (2019; 2020; 2023). For instance, world model based agents exhibit state-of-the-art performance on a wide range *single-agent* visual control tasks, such as Atari benchmark Bellemare et al. (2013), Deepmind Lab tasks Beattie et al. (2016) and Minecraft game Duncan (2011). These approaches typically involve two crucial components: (1) An encoder which process and compress environmental inputs (images, videos, text, control commands) into a more manageable format, such as a low-dimension latent representations and (2) a memory-augmented neural network, such as Recurrent Neural Networks (RNN) Yu et al. (2019), which equips agents with generalization capability. More importantly, compared to conventional planning algorithms that generate numerous rollouts to select the highest performing action sequence Bertsekas (2021), the differentiable world models can be more computationally efficient Levine & Koltun (2013); Wang et al. (2019); Zhu et al. (2020). Most recently, there are also efforts on developing a modularized world models for multi-agent environment, while requiring a separate large model Zhang et al. (2024) for inference or requiring assumptions on the value function decomposition Xu et al. (2022b).

**Communication in Multi-agent RL.** Recent works Jiang & Lu (2018); Foerster et al. (2016) adopt an end-to-end message-generation network to generate messages by encoding the past and current observation information. CommNet Sukhbaatar et al. (2016) aggregates all the agents' hidden states as the global message and shares the information among all agents or neighbors. MACI Pretorius et al. (2020) allows agents to share the their imagined trajectories to other agents through world model rollout. Furthermore, Kim et al. (2020) compresses the imagined trajectory into intention message to share with all other agents. To reduce the communication burden, ATOC Jiang & Lu (2018) and Liu et al. (2020) use the attention unit to select a group of collaborator to communicate while learning (or planning) directly in the (potentially high-dimensional) space. Egorov & Shpilman (2022) considers the notion of "locality" where the agent receive only history information from its neighbours in the environment. Ma et al. (2024) applies a static communication strategy by share the raw state information with nearby $k$-hop neighbors. In *CALL*, agent acquires both the latent state and latent intention information from other agents through prediction accuracy guided lightweight information sharing. Furthermore, we summarize the comparison between our work and related work in Table 2. Our work is also relevant to the literature on model-based RL, end-to-end autonomous driving and cooperative perception. Due to space limitation, we relegate the literature review to Appendix A.

## 2 *CALL* FOR EGO-CENTRIC MARL

**Basic Setting.** The distributed decision making problem in the multi-agent system is often cast as a (partial-observable) stochastic game $\langle \mathcal{S}, \{\mathcal{A}_i\}_{i\in\mathcal{N}}, P, \{r_i\}_{i\in\mathcal{N}}, \{\Omega_i\}_{i\in\mathcal{N}}, \gamma \rangle$, where $\mathcal{N}$ is the set of $N$ agents in the system, $\mathcal{S} \subseteq \mathbb{R}^{d_s}$ is the state space of the environment, and $\Omega_i$ and $\mathcal{A}_i$ are the observation space and action space for agent $i \in \mathcal{N}$, respectively Shapley (1953). Meanwhile, it is assumed that the state space is compact and the action space is finite. $\gamma \in [0, 1)$ is the discounting factor. It can be seen that from a single agent's perspective, each agent's decision making problem can be viewed as a Partial-Observable Markov Decision Process (POMDP) Kaelbling et al. (1998). At each time step $t$, each agent $i$ chooses an action $a_{i,t}$ by following policy $\pi_i : \mathcal{S} \to \mathcal{A}$, and denote the joint action by $\boldsymbol{a}_t = [a_{1,t} \cdots, a_{N,t}] \in \mathcal{A}$, $\mathcal{A} := \Pi_i \mathcal{A}_i$. Then the environment evolves from $s_t$ to $s_{t+1}$ following the state transition function $P(s_{t+1}|s_t, \boldsymbol{a}_t) : \mathcal{S} \times \mathcal{A} \times \mathcal{S} \to [0, 1]$. Each agent $i$ has a partial observation, e.g., sensory inputs of an autonomous vehicle, $o_{i,t} \in \Omega_i$ and receive the reward $r_{i,t} := r_i(s_t, \boldsymbol{a}_t)$.

*CALL* **for Ego-centric MARL.** We consider Ego-centric MARL setting Ozdaglar et al. (2021); Matignon et al. (2012); Zhang et al. (2021a; 2018), where each agent $i$ learns a policy $\pi_i$, $i \in \mathcal{N}$ aided by lightweight information sharing among agents. During the interaction with the environment, agent $i$ chooses an action $a_{i,t} \sim \pi_i$ based on received information $T_{i,t}$ and current state $x_{i,t}$. Then the goal of ego-centric learning for agent $i$ is to find a policy $\pi_i(\cdot|x_{i,t}, T_{i,t})$ that maximizes *her own value function* $v_i(x_{i,t}) \triangleq \mathbf{E}_{a_i \sim \pi_i}[Q_i^\pi(x_{i,t}, a_{i,t})]$, with Q-function $Q_i^\pi(x_{i,t}, a_{i,t}) = \mathbf{E}_{a_i \sim \pi_i}[\sum_t \gamma^t r_{i,t}]$ being the expected return when the action $a_{i,t}$ is chosen at state $x_{i,t}$.

As shown in Figure 1, each agent in *CALL* aims to train a world model $\mathtt{WM}_\phi$ to represent the latent dynamics of the environment and predict the reward $r$ and future latent state $z$. Each component is implemented as a neural network and $\phi$ is the combined parameter vector. Specifically, WM first learns a latent state $z_{i,t} \in \mathcal{Z} \subseteq \mathbb{R}^d$ based on the agent's partial observation from sensory inputs $o_{i,t}$ through autoencoding Kingma & Welling (2013).

Moreover, a RSSM Hafner et al. (2023; 2020) model is used to capture the context information of the current observation in the latent space by incorporating the hidden state in the encoder, i.e.,

Encoder: $z_{i,t} \sim q_\phi(z_{i,t}|h_{i,t}, o_{i,t}, T_{i,t})$,

For brevity, we denote the concatenation of $h_{i,t}$ and $z_{i,t}$ as the *model state* $x_{i,t} := [h_{i,t}, z_{i,t}] \in \mathcal{X}$. Then a recurrent model, e.g., RNN, uses the current model state to predict the next recurrent state $h_{i,t+1}$ given action $a_{i,t}$, i.e.,

Sequence Model: $h_{i,t+1} = f_\phi(x_{i,t}, a_{i,t}, T_{i,t})$,

where $h_{i,t+1}$ contains information about the latent representation for the next time step, i.e., $\hat{z}_{i,t+1} \sim p_\phi(\cdot|h_{i,t+1})$. Based on the model state, WM also predicts the reward $\hat{r}_{i,t+1} \sim p_\phi(\cdot|x_{i,t+1})$ and episode continuation flags $\hat{c}_{i,t+1} \sim p_\phi(\cdot|x_{i,t+1})$.

Figure 1: Illustration of *CALL*: Ego-centric learning in the two-agent case.

In what follows, we use an example with two interacting homogeneous agents, to illustrate the basic ideas in the proposed *CALL*; and this method is applicable to general heterogeneous multi-agent systems, as will be elaborated further below.

**An Illustration of *CALL* in A Two-Agent Case.** Figure 1 illustrates the interplay between two agents, where agent $i$, $i = 1, 2$, employs a world model each, in terms of latent state and latent intention $[z_i(t), h_i(t), w_i(t)]$, to represent its local dynamics model at time $t$. Since latent information has a very small memory footprint, agents can share $[z, h, w]$ via lightweight communications, i.e., blue arrows in Figure 1, enabling them to acquire information about other agents of interest and thereby alleviating the challenges of high-dimensionality and incomplete state information stemming from *partial observability*. Furthermore, from a single agent's perspective, non-stationarity would arise as agents adapt their action policy during the interaction. Fortunately, the generalization capabilities of WMs (particularly from the RNN) lend the agents the power of foresight, allowing them to better predict the future environment. By having access to the latent intention of neighboring agents, each agent can leverage the WM to reason about what to expect in the near future, thus mitigating the

challenge of *non-stationarity*. For instance, in the example in Figure 1, Agent 1 can use $[z_1, h_1, w_1]$ and $[z_2, h_2, w_2]$, together with the learned policy, to improve the prediction for 'near future' dynamics and obtain a more expanded view of its environment; and so can Agent 2.

**Notation.** The Frobenius norm of matrix $A$ is denoted by $\|A\|_F$ and the Euclidean norm of a vector is denoted by $|\cdot|$. A function $f : \mathbb{R}^n \to \mathbb{R}^m$ is said to be $L$-Lipschitz, $L > 0$, if $|f(a) - f(b)| \le L|a - b|$, $\forall a, b \in \mathbb{R}^n$. $\mathbf{E}[X]$ is the expected value of random variable $X$.

## 3 PREDICTION ERROR AND SUB-OPTIMALITY GAP IN *CALL*

*CALL* benefits from the innovative synergy of WM's generalization capability and lightweight information sharing to improve the performance of ego-centric MARL. To develop a systematic understanding, in this section we first quantify the prediction error (or uncertainty) when using WMs for predictive rollouts, and investigate the impact of the insufficient information through a structural dissection of the prediction error. More importantly, the prediction error analysis offers valuable insights on how to synergize WM's generalization and information sharing to improve the prediction in *CALL*. We also quantify the benefits of the proposed information sharing scheme in *CALL* on the learning performance by deriving the upper bound of the sub-optimality gap.

### 3.1 ERROR ANALYSIS OF MULTI-STEP PREDICTION

**RNN Model.** At each time step $t$, the agent will leverage the sequence model $f_\phi$ in the world model to generate imaginary trajectories $\{\hat{z}_{i,t+k}, a_{i,t+k}\}_{k=1}^K$ with rollout horizon $K > 0$, based on the model state $x_{i,t} := [z_{i,t}, h_{i,t}]$, policy $a_{i,t} \sim \pi_i$ and shared information $I_{i,t}$. We consider the sequence model in the world model to be the RNN, which computes the hidden states $h_{i,t}$ and state presentation $z_{i,t}$ as follows,

$$h_{i,t+1} = f_h(x_{i,t}, a_{i,t}, I_{i,t}), \quad z_{i,t+1} = f_z(h_{i,t+1}), \tag{1}$$

where $f_h$ maps the input to the hidden state and $f_z$ maps the hidden state to the state representation. In our theoretical analysis, for simplicity, we abuse notation slightly by using $x_{i,t}$ to represent the 'updated' model state after incorporating the shared information $I_{i,t}$ when no confusion may arise. Following the same line as in previous works Lim et al. (2021); Wu et al. (2021), we consider $f_h = Ax_{i,t} + \sigma_h(Wx_{i,t} + Ua_{i,t} + b)$ and $f_z = \sigma_z(Vh_{i,t+1})$, where $\sigma_h$ is a $L_h$-Lipschitz element-wise activation function (e.g., ReLU Agarap (2018)) and $\sigma_z$ is the $L_z$-Lipschitz activation functions for the state representation. The matrices $A, W, U, V, b$ are trainable parameters.

Without loss of generality, we have the following standard assumptions on actions and RNN model.

**Assumption 1** (Action and Policy). *The action input is upper bounded, i.e., $|a_{i,t}| \le B_a$, $t = 1, \cdots$ and the policy $\pi_i$ is $L_a$-Lipschitz for all $i \in \mathcal{N}$, i.e., $d_X(\pi_i(\cdot|x) - \pi_i(\cdot|x')) \le L_a d_X(x, x')$, $x, x' \in \mathcal{X}$, where $d_A$ and $d_X$ are the corresponding distance metrics defined in the action space and state space.*

**Assumption 2** (Weight Matrices). *The Frobenius norms of weight matrices $W$, $U$ and $V$ are upper bounded by $B_W$, $B_U$ and $B_V$, respectively.*

Assumption 1 and Assumption 2 also imply that the world model state $x_{i,t}$ is bounded and we assume $|x_{i,t}| \le B_x$. Both assumptions are standard assumptions in the analysis of RNN Lim et al. (2021); Wu et al. (2021); Pan & Wang (2011). In particular, the Lipschitz assumptions on the policy is widely used in the literature on MDPs analysis Shah & Xie (2018); Dufour & Prieto-Rumeau (2013; 2012). An example of the policy that satisfies Assumption 1 is the linear controller as considered in the world model proposed in Ha & Schmidhuber (2018).

**Structure of Multi-step Lookahead Prediction Error.** The prediction error is the difference between the underlying true state and the state predicted by RNN. Specifically, the prediction error at prediction steps $k \ge 1$ is defined as follows,

$$\epsilon_{i,t+k} = z_{i,t+k} - \hat{z}_{i,t+k} := (z_{i,t+k} - \bar{z}_{i,t+k}) + (\bar{z}_{i,t+k} - \hat{z}_{i,t+k}), \tag{2}$$

where $z_{i,t+k}$ is the state representation for the ground truth state that is aligned with the ego agent's planning objective (e.g., the planning horizon $K$). $\hat{z}_{i,t+k}$ is the prediction generated by RNN (ref. Eqn. equation 1) when using the agent's local information, i.e., $x_{i,t}$. Meanwhile, we denote $\bar{z}_{t+k}$ as the prediction generated by RNN if the shared information from other agents is employed as input. To further analyze the impact of information sharing, we decompose the prediction error into two parts in Equation (2). The first term captures the *generalization error* inherent in the RNN, while the second

term pertains to the *epistemic error*, arising from the absence of information sharing, For our analysis, we assume the RNN is trained with supervised learning on $n$ i.i.d. samples of state-action-state sequence and the empirical loss is $l_n$. Meanwhile, let the expected total-variation distance between the true state transition probability $P(z'|z, a)$ and the predicted one $\hat{P}(z'|z, a)$ be upper bounded by $\mathcal{E}_P$, i.e., $\mathbf{E}_\pi[D_{\mathrm{TV}}(P||\hat{P})] \leq \mathcal{E}_P$. Moreover, we denote the gap of the input model state at time step $t$ as a random variable $\epsilon_x$ with expectation $\mathcal{E}_x := \arg\max_k \mathbf{E}[x_{t+k} - x_{i,t+k}]$, where $x_t$ is the model state obtained by using shared information.

For brevity, we denote $M = B_V B_U \frac{(B_W)^k - 1}{B_W - 1}$, $\Psi_k(\delta, n) = l_n + 3\sqrt{\frac{\log\left(\frac{2}{\delta}\right)}{2n}} + \mathcal{O}\left(d\frac{MB_a(1 + \sqrt{2\log(2)k})}{\sqrt{n}}\right)$, where $d$ is the dimension of the latent state representation and $N_1 = L_h L_z L_a UV$, $N_2 = L_h L_z VW + L_z VA$. Then we obtain the following result on the upper bound of the prediction error.

**Theorem 1.** *Given Assumptions 1 and 2 hold, with probability at least $1 - \delta$, we have the multi-step lookahead prediction error $\epsilon_{i,t+k}$, for $k \geq 1$, is upper bounded by*

$$\epsilon_{i,t+k} \leq \sum_{j=1}^{k} N_1^j \left(\sqrt{\Psi_h(\delta, n)} + 1/\delta(N_2\mathcal{E}_x + 2hB_x\mathcal{E}_P)\right) := \mathcal{E}_{\delta,k}$$

The upper bound in Theorem 1 reveals the impact of the prediction horizon $k$, the generalization error of RNN (the first term), model state error $\mathcal{E}_x$ (the second term), and modeling error $\mathcal{E}_P$ (the third term), thereby providing the guidance on what information is essential in order to reduce the prediction error. In particular, the generalization error of RNN stems from the training process and is related to the training loss and the number of training samples. In general, it can be seen that as the prediction horizon $k$ increases, the modeling error and generalization error in the upper bound tends to have more pronounced impact on the overall prediction. Meanwhile, the summation structure of the upper bound also implies potential error accumulation over prediction horizons. Guided by the insights from Theorem 1, we next elaborate how to synergize the WM's generalization with information sharing to improve the prediction in *CALL*.

**Prediction performance gain from information sharing.** The error term $\mathcal{E}_x$ originates from the gap between model states $x_t$ and $x_{i,t}$, where the latter is obtained by using ego-agent's local observation and shared information. To this end, in the proposed *CALL*, ego-agent will benefit from accessing other agent's local state information, i.e., the model state $\{z_{j,t}, h_{j,t}\}, j \in \mathcal{G}_t \subseteq \mathcal{N}$, to acquire a better estimation of the model state $x_t$ from a subset of agents $\mathcal{G}_t$. Meanwhile, the error term $\mathcal{E}_P$ quantifies the disparity of the state transition model predicted by RNN and the underlying real transition model. To alleviate the modeling error and the curse of non-stationarity, it is plausible for the agents to share their intentions $w_{j,t}$ when needed. More concretely, assume that after acquiring the information $\{z_{j,t}, h_{j,t}, w_{j,t}\}, j \in \mathcal{G}_t \subseteq \mathcal{N}$, the error terms $\mathcal{E}_x$ and $\mathcal{E}_P$ are reduced by $\varepsilon_x, \varepsilon_p$, respectively, then we can obtain that the prediction error can be improved by at least $\sum_{j=1}^{k} N_1^j 1/\delta(N_2\varepsilon_x + 2hB_x\varepsilon_P)$ (ref. Theorem 1).

### 3.2 SUB-OPTIMALITY GAP

Next, we carry out theoretical studies to quantify the benefits of the information sharing in *CALL*. Notably, the conventional solution concept in seeking equilibrium in partially observable stochastic games (POSGs) may not work well with the real-time applications such as autonomous driving. For instance, previous theoretical results on equilibrium-based solutions have primarily focused highly structured problems such as two-player zero-sum game Kozuno et al. (2021); Zinkevich et al. (2007) or potential game Mguni et al. (2021); Yang & Wang (2020). However, the complex real-world tasks often deviate from these settings and it also has been shown that equilibrium computation is PPAD in general stochastic games Daskalakis et al. (2009). In light of these well-known computational and statistical hardness results Jin et al. (2020), we instead advocate to reap the benefits of information sharing by evaluating the learning performance gain due to the shared information, akin to natural learning algorithms Ozdaglar et al. (2021).

We provide the upper bound on the prediction error in Theorem 1, from which we identify the information needed to reduce the prediction error while addressing the partial observability and non-stationarity in distributed RL. Clearly, the prediction error has direct impact on the agent's decision making performance. We characterize the condition on the prediction error under which the sub-optimality gap can be upper bounded by a desired level. We first impose the following assumptions on the reward.

**Assumption 3.** *The one step reward $r(x, \boldsymbol{a})$ is $L_r$-Lipschitz, i.e., for all $x, x' \in \mathcal{X}$ and $\boldsymbol{a}, \boldsymbol{a}' \in \mathcal{A}$, we have,*

$$|r(x, \boldsymbol{a}) - r(x, \boldsymbol{a}')| \leq L_r(d_X(x, x') + d_A(a, a')),$$

*where $d_X$ and $d_A$ are the corresponding metrics in the state space and action space, respectively.*

The assumption on the reward function in Assumption 3 follows the same line as (or less restrictive than) the ones in the literature Shah & Xie (2018); Dufour & Prieto-Rumeau (2015); Chow & Tsitsiklis (1991); Rust (1997). Notably, the reward function considered in this assumption is defined on the latent state $x$, which can be obtained by encoding the state $s$ using the encoder. Let $\bar{E} := \frac{1-\gamma^{K-1}}{1-\gamma} L_r(1 + L_\pi) + \gamma^K L_Q(1 + L_\pi)$, $\mathcal{E}_{\max} = \max_t \mathcal{E}_{\delta,t}$ and $L_Q = L_r/(1 - \gamma)$, then we obtain the following proposition on the impact of the prediction error on the sub-optimality gap, i.e., the gap between the optimal value function $v_i^*(x_{i,t}|T_{i,t})$ and $v_i^*(x_{i,t}|I_t)$, where $I_t$ contains all the information in the system and $T_{i,t}$ is the information shared to agent $i$ at time $t$.

**Proposition 1.** *Given Assumption 3 holds. If the prediction error induced by using shared information $T_{i,t}$) satisfies $\mathcal{E}_{\max} \leq \epsilon/\bar{E}$, then the sub-optimality gap is upper bounded by $\epsilon$, i.e., $|v_i^*(x_{i,t}|T_{i,t}) - v_i^*(x_{i,t}|I_t)| \leq \epsilon$.*

Proposition 1 shows the connection between the world model's prediction error and the sub-optimality gap of the value function. As expected, to reduce the sub-optimality gap, it is imperative to reduce the prediction error. Notably, if the prediction error induced by information $T_{i,t}$) is small enough, $T_{i,t}$) can be seen as "locally sufficient information" for achieving desired prediction accuracy. As an example, for driving, it suffices for an agent to acquire information about vehicles within its local view and its planned path. Meanwhile, thanks to the latent representations in world models, obtaining such information only requires lightweight communication among agents, which is highly desired in the practical distributed RL implementations. The theoretical results in Theorem 1 and Proposition 1 lay the foundation for our proposed *CALL*. The proof can be found in Appendices B and C.

### 3.3 PREDICTION-ACCURACY-DRIVEN INFORMATION SHARING

*CALL* allows agents to adaptively adjust their information sharing based on real-time evaluation of their prediction accuracy. By leveraging insights from our theoretical results in Theorem 1 and Proposition 1, we understand how prediction errors accumulate over time due to factors like partial observability and non-stationarity, which in turn widen the sub-optimality gap. *CALL* continuously monitors these errors, enabling agents to detect when their world models are underperforming and triggering selective, lightweight information sharing to correct course.

We summarized the proposed *CALL* in Algorithm 1. Specifically, each agent begins by encoding its local sensory inputs $o_{i,t}$ and planned actions into compact latent representations, specifically latent state $z_{i,t}$ and latent intention $w_{i,t}$. This encoding facilitates lightweight communication. At each time step, agent $i$ evaluates its prediction errors by comparing the previously predicted latent states and intentions $\hat{X}_{i,t} = \{\hat{z}_{i,t-k}, \hat{w}_{i,t-k}\}_{k=0}^{K-1}$ against the actual observations $X_{i,t} = \{z_{i,t-k}, w_{i,t-k}\}_{k=0}^{K-1}$ from the last $K$ time steps. The prediction error is then calculated as: $\mathcal{E}_{i,t} = \|\hat{X}_{i,t} - X_{i,t}\|$. If this error exceeds a predefined threshold $c$, the agent increase its communication range $\mathcal{G}_{i,t}$ (e.g., increase by 5 meters) and exchanges relevant latent information. This ensures that only lightweight and targeted information sharing takes place, reducing unnecessary communication overhead.

Each agent continuously updates its policy by computing its value function $v_i$ based on the shared information and world model predictions for the next $K$ steps. This adaptive process allows agents to refine their decision-making, improving their ability to handle partial observability and non-stationarity in complex environments. By monitoring prediction errors in real time and engaging in selective information sharing, *CALL* can achieve scalable and efficient performance in ego-centric MARL systems. To assess the efficiency and practical viability of *CALL*, we next conduct extensive experiments and ablation studies.

## 4 EXPERIMENTS

*Environment Settings.* We validate the efficiency of *CALL* in CARLA, an open-source simulator with high-fidelity 3D environment Dosovitskiy et al. (2017). At time step $t$, agent $i$, $i \in \mathcal{N}$ receives bird-eye-view (BEV) as observation $o_{i,t}$, which unifies the multi-modal information Liu et al. (2023);

---

**Algorithm 1** *CALL* for Ego-centric MARL.

---

**Require:** World Model $\mathtt{WM}_\phi$, initial policy $\pi_{i,0}$, all agents $\mathcal{N}$, agents in the initial communication range $\mathcal{G}_0 \subseteq \mathcal{N}$, planning horizon $K$, prediction accuracy threshold $c$.

   **for** each agent $i$, step $t = 1, 2, \cdots$ **do**

      Encode local sensory input $o_{i,t}$ and planned actions $\{a_{i,t}\}_t^{t+K}$ and obtain latent representation $\{z_{i,t}, h_{i,t}, w_{i,t}\}$.

      Evaluate prediction error (Theorem 1) and update communication range $\mathcal{G}_t$ accordingly

      # Prediction accuracy guided lightweight information sharing

      Exchange information $T_{i,t} = \{z_{j,t}, h_{j,t}, w_{j,t}\}$ with selected agents $j \in \mathcal{G}_t$

      **for** $k = 1, 2, \cdots, K$ **do**

         Predict $\hat{z}_{i,t+k}, \hat{h}_{i,t+k}$ using $\mathtt{WM}_\phi(o_{i,t}, T_{i,t}, \pi_{i,t})$.

         # Harness WM's generalization capability

      **end for**

      Compute $v_i(x_{i,t}|T_{i,t})$, e.g., Equation (3).

      Update policy: $\pi_i^t \leftarrow \arg\max v_i$.

   **end for**

---

Li et al. (2023). Furthermore, by following the planned waypoints generated by the CARLA planning module, agents navigate the environment by executing commands $a_{i,t}$ and receive the reward $r_{i,t}$ from the environment. We define the reward as the weight sum of five attributes: safety, comfort, driving time, velocity and distance to the waypoint. The details of the CARLA environment and reward design are relegated to Appendix E. We also include the detailed discussion on baseline and benchmark in Appendix E.3 and the impact of the threshold $c$ in Appendix G.

Notably, we consider with two configurations: I) 150 agents and II) 230 agents, which requires more challenging maneuver. Due to space limitation, the experiment results for the latter configuration are relegated to Appendix E.4. In both cases, our findings are consistent and corroborate that the proposed *CALL* exhibits great potential to navigate in complex environments. In our figures, we use shaded area to represent the standard deviation.

*Trajectory Planning Tasks.* In our experiments, we consider the trajectory planning tasks in autonomous driving, where the objective of the ego vehicle is to safely navigate through the traffic to reach the designed exit point. Each agent first learns the target waypoints for the vehicle to drive to and then, based on these higher-level decisions, makes the lower-level decisions, such as steering angle, throttle control, and brake control in order to reach the waypoints Hu et al. (2018); Naveed et al. (2021); Lu et al. (2023). To distinguish the difference, we use notation $a_{i,t}$ to represent the action of agent $i$ at time step $t$ and $w_{i,t}$ to represent a set of waypoints planned at time step $t$. Note that the waypoints will remain fixed until the path is re-planned at $(t + K)$-time steps. In our experiments, we adopt the *CALL* for the lower-level of decision making while adhering to the planned waypoints. Specifically, the agent aims to find the next $K$-step actions such that the value function $v_i$ is maximized. For convenience, let $\hat{x}_{i,t+k} := [\hat{z}_{i,t+k}, \hat{h}_{i,t+k}]$ and $\hat{r}_{i,t} := r_i(\hat{x}_{i,t}, a_{i,t})$, then the value function is defined as,

$$v_i(x_{i,t}) = \mathbf{E}_{\pi_i}\left[\sum_{k=0}^{K-1} \gamma^k \hat{r}_{i,t+k} + \gamma^K v_i(\hat{x}_{i,t+K})\right], \tag{3}$$
$$\pi_i \leftarrow \arg\max\nolimits_{\{a_{i,t+k}, k=0,\cdots,K-1\}} v_i(x_{i,t}).$$

**Ego-centric MARL Performance with *CALL*.** We summarize the proposed *CALL* in Algorithm 1 (see Appendix E), where the outer-loop $t = 1, 2, \cdots$ represents agent's interaction with the environment and the inner-loop $k = 1, \cdots, K$ is the world model's rollout horizon. For ease of exposition, we focus on the setting where agents share the same encoder-decoder architecture so that the agent is able to decode the shared model state $[z, h]$ directly. We will elaborate the case with heterogeneous agents in Appendix F. We build the world model upon Dreamer V3 Hafner et al. (2023) and the details of the model training are summarized in Appendix E. During training, the agent selects an communication range $\mathcal{G}_t$ and exchange information with agents within that range to acquire their model state and intention $\{x_{j,t}, w_{j,t}\}$, $j \in \mathcal{G}_t \subseteq \mathcal{N}$. The shared information is integrated into agent's 'updated' BEV for decision making. During the interaction, the agent adaptively update the communication range to achieve prediction accuracy guided information sharing.

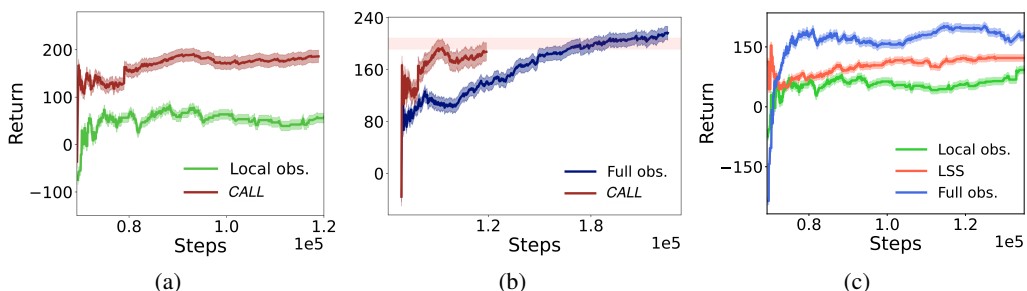

(a)         (b)         (c)

Figure 2: (a) RL performance comparison between two settings: Local observation only (no communication) vs. *CALL* (sharing latent state & intention, i.e., waypoints). (b) RL performance comparison: *CALL* vs. full observation. (c) Ablation studies on the impact of latent state sharing (LSS): Full observation, 'latent state sharing (LSS)', and local observation only.

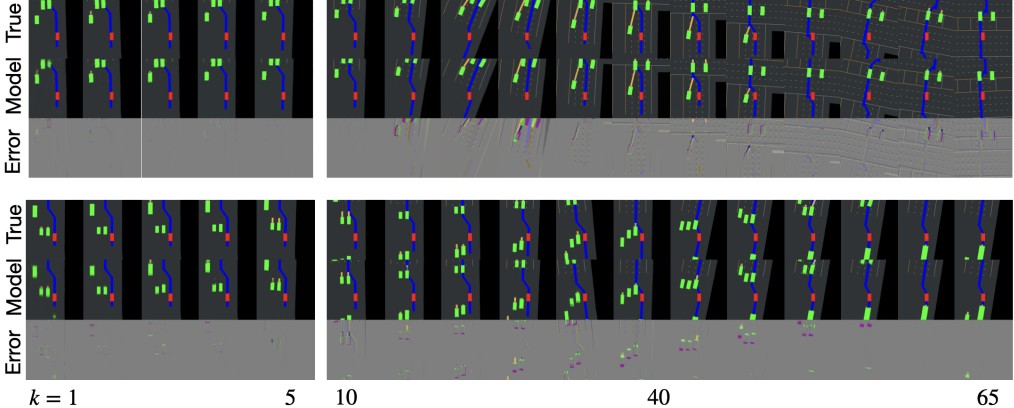

Figure 3: Multi-step predictions results. The first five frames are used as context input; and the model predicts the future frames (the second row); the third row is the error. The yellow line in front of the vehicles is the waypoints. The first row is the results with local observation only, while the second row is the prediction results by synergizing WM's generalization capability with lightweight information sharing in *CALL*. Additional results on WM's generalization capability in *CALL* can be found in Appendix E.

As shown in Figure 2(a), we first compare the learning performance in two settings: one based on the shared information, and another relying on local observation alone (as in DreamerV3 Hafner et al. (2023) and Think2Drive Li et al. (2024)). As expected, the ego agent using shared information achieves significant around $100\%$ performance improvement. Next, we compare the learning performance with full observation case in Figure 2(b), and it is somewhat surprising to observe that the training using *CALL* can in fact result in faster learning compared to the case with full observation (i.e, all available observations). As shown in Figure 2(b), the training curve for the full observation setting reaches the return value around 200 at step 180k; in contrast, the agent with shared information achieves this value at step 120k. Our intuition is that the full observation may contain noisy and non-essential information for agent's decision making, which can impede the learning speed. Moreover, the performance gain in *CALL* incurs very low communication overhead. Specifically, we show the bandwidth requirements in Figure 5 (Appendix D). In average, it requires significantly lower data transmission (0.106 MB) than the case of sharing information with all agents, which requires more than 5.417 MB data in a 230 vehicle system per 0.1 second.

In what next, we conduct ablation studies to show the benefits of *CALL* on addressing partial observability and non-stationarity issues in ego-centric MARL, respectively.

**Ablation Studies.** 1) *Addressing Partial Observability*. We elucidate the impact of the model state sharing on the learning performance through both the learning performance comparison (Figure 2(c)) and an illustrative example (Figure 4(c)). As shown in Figure 4(c), by integrating the shared model

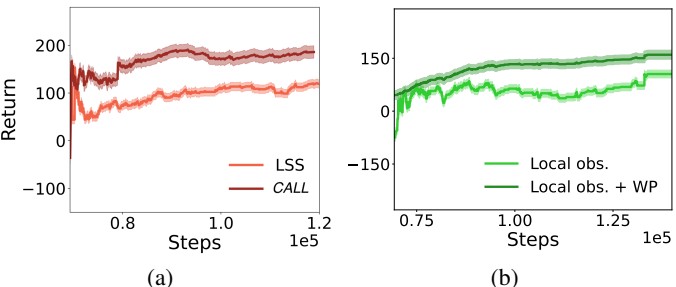 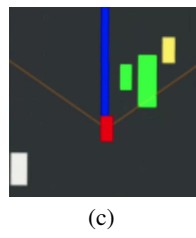

(a)    (b)    (c)

Figure 4: (a-b) Ablation studies on the impact of waypoints: 'Local obs. + WP' represents the case where the waypoints information is utilized together with local observation. (c) Illustration on ego vehicle's (red) observability. The green vehicles can be observed directly. The yellow vehicle can be 'observed' by decoding the middle vehicle's model state. The gray vehicle is not observed.

state, the agent is able to acquire the essential information that are beyond its own sensing limitations. For instance, even the yellow vehicle is at ego vehicle's blind spot but its location is critical for ego agent's decision making along the planned waypoints (the blue line). Furthermore, as evidenced in Figure 2(c), sharing the model state brings promising performance gain comparing with the scenario without communication. More demonstrations on the BEV can be found in Figure 14.

2) *Addressing Non-stationarity*. To evaluate the prediction performance, we first compare the BEV prediction results under three settings: with information sharing, without information sharing and full observation in Appendix E.5. Consistent with Theorem 1, it is evident that the prediction error increases with the rollout horizon in all cases. Meanwhile, it can be seen that *CALL* can greatly improve the BEV prediction and hence benefit the planning. For instance, as shown in Figure 3, by synergizing WM's generalization capability and lightweight information sharing, *CALL* in generally can make better prediction. On the other hand, among the shared information, waypoints encapsulate agents' intention in the near future and is particularly crucial for better prediction in the non-stationary environment. In this regard, we quantify the benefits of sharing waypoints by studying the overall learning performance gain. As can been seen in Figure 4(a), sharing waypoints information results in around $100\%$ performance gain comparing with the case with latent state sharing ("LSS"). Meanwhile, Figure 4(b) demonstrates that the waypoints information can greatly help with the learning even in the case when agents only have have access to their own observation of the environment. The results in Figure 4(a) and Figure 4(b) show that the synergy between WM generalization capability and the information sharing (especially the intention sharing) in *CALL* is the key to mitigate the poor prediction challenge in the distributed RL.

## 5    CONCLUSION

In this work, we introduce *CALL* to address the key challenges of partial observability and non-stationarity in ego-centric MARL in complex, high-dimensional environments. The core innovation of *CALL* lies in the synergization of the generalization capability of world models and lightweight information sharing. In particular, *CALL* facilitates prediction-accuracy-driven information sharing, which allows agents to selectively and flexibly exchange only the most relevant information, improving prediction accuracy while keeping the approach efficient and suitable for real-time decision-making. Our theoretical results in Theorem 1 and Proposition 1 demonstrate how world models can improve prediction performance and reduce the sub-optimality gap through the use of shared information. Extensive experiments in autonomous driving tasks show that *CALL* achieves promising results, with ablation studies supporting our theoretical analysis. Looking ahead, we hope *CALL* can open a new avenue to devise practical *CALL* algorithms in other applications involving planning and navigation. Meanwhile, the consideration on the uncertainty during the information sharing is also worth to explore. Another important direction is exploring privacy-preserving mechanisms for information sharing, ensuring that agents can collaborate without revealing sensitive or private information.

## ETHICS STATEMENT

In developing *CALL*, we are mindful of the ethical implications surrounding multi-agent reinforcement learning, particularly in high-stakes applications such as autonomous driving. *CALL* is designed to enhance decision-making efficiency and scalability, but it is crucial to ensure that this technology is applied responsibly. We commit to considering the safety, fairness, and privacy of all individuals impacted by the deployment of multi-agent systems. Furthermore, our approach involves the sharing of information between agents, which must be handled with strict adherence to data privacy and security standards. Ensuring that autonomous systems behave safely and fairly, without bias or unintended consequences, remains a priority in the development and deployment of *CALL*.

## REPRODUCIBILITY STATEMENT

To ensure the reproducibility of our results, we provide a detailed description of the *CALL* algorithm, including the source code and all experimental settings, in the main text and supplementary materials. Hyperparameters, network architectures, and training protocols are described in full, and all data sets and simulated environments used in our experiments will be made publicly available upon publication, allowing other researchers to replicate and build upon our findings with ease.

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

# Appendix.

## A   RELATED WORK

**Model-based MARL.** Model based RL in single-agent setting has shown promising results in both theoretical analysis and practical experiments, especially in terms of the sampling efficiency Moerland et al. (2023); Yarats et al. (2021); Kaiser et al. (2019); Janner et al. (2019). However, the studies on model-based MARL has just recently started to attract attention. For instance, Zhang et al. (2018) investigates a two-player discounted zero-sum Markov games and establishes the sample complexity of model-based MARL. Park et al. (2019) proposes a RNN based actor-critic networks and policy gradient method to promote agents cooperation by sharing the gradient flows over the agents during the centralized training. Zhang et al. (2021b) utilizes opponent-wise rollout policy optimization in MARL, where the ego agent models all other agents during the decision-making process. Xu et al. (2022b) suggests using world model rollout in cooperative MARL but requires the access of all agent's history information. Xu et al. (2022b) proposed to use the world model rollout in cooperative MARL while requires the access of all agent's history information. Furthermore, Chockalingam et al. (2018) extends the world model for MARL by defining a meta-controller that takes all agents state information as input to generate the teams control actions. In an open multi-agent system, where the number of the agents changes over time, the aforementioned methods may suffer from the scalability and stability issue.

**End-to-end Autonomous Driving.** The field of end-to-end systems has gained a lot popularity due to the availability of large-scale datasets and closed-loop evaluation Chen et al. (2023). In particular, our work is relevant to the world model (model-based) learning paradigm. Clearly, modeling the complex world dynamics plays an important role on the learning performance while also poses significant challenges. In this regard, Chen et al. (2021) introduces a probabilistic sequential latent environment model to address the issues on high dimensnionality and partial observability by utilizing the latent representation and historical observations. Recent studies, as demonstrated by Wang & Meger (2023), have revealed the difficulty of learning holistic models in environments with non-stationary components. This observation has prompted investigations into modular representations to effectively decouple world models into distinct modules. For instance, Wang & Meger (2023) considers three components when training the model, i.e., action-conditioned, action-free and static. In Pan et al. (2022), the dynamics model is decoupled into passive and active components. Notably, such training methods generally require the access to the full observation of the world for the disentanglement. In the contrary, the world model training in the proposed `WM-DRL` framework considers the setting of ego agent's partial observability in the multi-agent system. A more thorough review can be found in Chen et al. (2023).

**Cooperative Perception in Autonomous Driving.** Cooperative perception (CP) seeks to extend single vehicle's perception range by the exchange of local sensor data with other vehicles or infrastructures Kim et al. (2015). In the transportation system studies, CP has been widely used for 3D object detection by using LiDAR point cloud Chen et al. (2019b), camera images Arnold et al. (2020) and/or RADAR Rauch et al. (2012). However, sharing massive amounts of raw data among vehicles can be prohibited in practice. Additionally, the processing of those high-volume data will introduce extra latency Yang et al. (2021). To this end, Xu et al. (2022a) presents mobility-aware sensor scheduling algorithm, considering both viewpoints and communication quality, to efficiently schedule cooperative vehicles for the most beneficial data exchange. Our proposed `WM-DRL` framework goes beyond the CP tasks and aims to address the decision making of ego vehicles in the multi-agent system by taking advantage of the light-weight communication.

**BEV Representation in Autonomous Driving.** Learning the world model directly in raw image space is challenging and may not suitable for autonomous driving. This approach is prone to missing crucial small details, such as traffic lights, in the predicted images Chen et al. (2023). Meanwhile, the autonomous driving systems generally equip with diverse sensors with different modalities, which makes the sensor fusion to be essential and a standard approach in practice. For instance, Liu et al. (2023) proposes BEVFusion unifies multi-modal features in the shared BEV representation space. Fadadu et al. (2022) shows that the fusion of sensor data in a unified BEV can improve the perception and prediction in autonomous driving. Zhang et al. (2021c) trains an end-to-end RL expert that maps BEV images to continuous low-level actions for imitation learning. Similarly, Chen et al. (2020)

Table 3: Summary of Notations

| Notation | Description |
| --- | --- |
| $\mathcal{N}$ | Set of $N$ agents in the system |
| $\mathcal{S} \subseteq \mathbb{R}^{d_s}$ | State space of the environment |
| $\Omega_i$ | Observation space for agent $i$ |
| $\mathcal{A}_i$ | Action space for agent $i$ |
| $\gamma \in [0, 1)$ | Discount factor |
| $a_{i,t}$ | Action chosen by agent $i$ at time $t$ |
| $\boldsymbol{a}_t$ | Joint action $[a_{1,t}, \cdots, a_{N,t}]$ |
| $o_{i,t} \in \Omega_i$ | Local observation of agent $i$ at time $t$ |
| $r_{i,t}$ | Reward received by agent $i$ at time $t$ |
| $\pi_i$ | Policy of agent $i$ |
| $z_{i,t} \in \mathcal{Z}$ | Latent state representation |
| $h_{i,t}$ | Hidden state from RNN |
| $x_{i,t} = [h_{i,t}, z_{i,t}]$ | Model state |
| $w_{i,t}$ | Latent intention (planned waypoints) |
| $\mathcal{G}_t$ | Set of agents in communication range at time $t$ |
| $T_{i,t}$ | Information shared with agent $i$ at time $t$ |
| $K$ | Prediction/planning horizon |
| $\mathcal{E}_P$ | Upper bound on expected total-variation distance |
| $\mathcal{E}_x$ | Expected gap in model state |
| $L_h, L_z$ | Lipschitz constants for activation functions |
| $B_W, B_U, B_V$ | Upper bounds on weight matrices norms |
| $B_a, B_x$ | Upper bounds on action and state norms |
| $L_a, L_r$ | Lipschitz constants for policy and reward |
| $c$ | Prediction accuracy threshold |

use the BEV as the privileged information in order to obtain a privileged agent as a teaching agent. Chen et al. (2019a) leverages the latent representation of BEV as state to train a model-free RL agent. Bansal et al. (2018) also uses the BEV as the training input.

## B  PROOF OF THEOREM 1

**Table of Notations.** In Table 3, we summarize the notations used.

**Structure Dissection of Prediction Error.** We define the prediction error to be the difference between the underlying true state and the state predicted by RNN. In particular, we consider the structure dissection of the prediction as follows: at prediction steps $k = 1, \cdots, K$,

$$
\begin{aligned}
\epsilon_{i,t+k} &= z_{i,t+k} - \hat{z}_{i,t+k} \\
&:= \underbrace{(z_{i,t+k} - \bar{z}_{i,t+k})}_{\text{Generalization Error}} + \underbrace{(\bar{z}_{i,t+k} - \hat{z}_{i,t+k})}_{\text{Epistemic Error}}.
\end{aligned}
\tag{4}
$$

For clarity, we summarize the notations in the list below.

- $z_{i,t+k}$: state representation for the ground truth state that is aligned with agent's individual planning objective (e.g., the planning horizon $K$).

- $\hat{z}_{i,t+k}$: the prediction generated by RNN (ref. Eqn. equation 1) when using the agent's local observation, i.e., $x_{i,t}$.

- $\bar{z}_{t+k}$: the prediction generated by RNN if the Locally Sufficient Information (LSI) is employed as input.

To further analyze the impact of LSI, we decompose the prediction error into two parts: (1) Generalization error inherent in the RNN and (2) Epistemic error, arising from the absence of LSI. Next, we first investigate the generalization term.

**Generalization Error Term.** We consider the setting where RNN model is obtained by training on $n$ i.i.d. samples of state-action-state sequence $\{x_t, a_t, x_{t+1}\}$ and the empirical loss is $l_n$ with loss function $f$. The RNN is trained to map the one step input, i.e., $x_t, a_t$, to the output $x_{t+1}$. Particularly, the world model leverage the RNN to make prediction over the future steps. In the analysis of the first term in Equation (4), with the input satisfying LSI, the error term capture the generalization of using RNN model on a new state-action-state sample $\{x_t, a_t, x_{t+1}\}$. Following the standard probably approximately correct (PAC) learning analysis framework, we first recall the following results Alnajdi et al. (2023); Wu et al. (2021) on the RNN generalization error. For brevity, we denote

$M = B_V B_U \frac{(B_W)^k - 1}{B_W - 1}$, $\Psi_k(\delta, n) = l_s + 3\sqrt{\frac{\log\left(\frac{2}{\delta}\right)}{2n}} + \mathcal{O}\left(d\frac{MB_a(1 + \sqrt{2\log(2)k})}{\sqrt{n}}\right)$, where $d$ is the dimension of the latent state representation

**Lemma 1** (Generalization Error of RNN). *Assume the weight matrices satisfy Assumption 2 and the input satisfies Assumption 1. Assume the training and testing datasets are drawn from the same distribution. Then with probability at least $1 - \sigma$, the generalization error in terms of the expected loss function has the upper bound as follows,*

$$\mathbf{E}[f(z_{i,t+k} - \bar{z}_{i,t+k}] \leq l_n + 3\sqrt{\frac{\log\left(\frac{2}{\delta}\right)}{2n}} + \mathcal{O}\left(L_r d_y \frac{dMB_a(1 + \sqrt{2\log(2)k})}{\sqrt{n}}\right)$$

In particular, the results in Lemma 1 considers the least square loss function and the generalization bound only applies to the case when the data distribution remains the same during the testing. In our case, since the testing and training sets are collected from the same simulation platform thus following the same dynamics. For simplicity, in our problem setting, we assume the underlying distribution of input $\{x_t, a_t\}$ is assumed to be uniform. Subsequently, we establish the upper bound for the generalization error. Let $\epsilon_{t+k} = z_{i,t+k} - \bar{z}_{i,t+k}$, then we have, with probability at least $1 - \delta$,

$$\epsilon_{t+k}^{\text{RNN}} \leq \sqrt{\Psi_k(\delta, n)} \tag{5}$$

**Epistemic Error Terms.** The epistemic error term stems from agent's lack of LSI. In the multi-agent system, the agents knowledge can be further decomposed into stationary part and non-stationary part. Without loss of generality, we assume $\bar{z}_{i,t} = [\bar{z}_{i,t}^s, \mathbf{0}]^\top + [\mathbf{0}, \bar{z}_{i,t}^{ns}]^\top$, where $\mathbf{0}$ is the all zero vector with proper dimension. Then we have the epistemic error with the following form.

- At current time step $t$:

$$\epsilon_t^{\text{Epistemic}} := \bar{z}_{i,t} - \hat{z}_{i,t}$$
$$\triangleq \underbrace{\bar{z}_{i,t}^s - \hat{z}_{i,t}^s}_{\text{Stationary}} + \underbrace{\bar{z}_{i,t}^{ns} - \hat{z}_{i,t}^{ns}}_{\text{Non-stationary}}$$
$$= \underbrace{\bar{z}_{i,t}^s - \hat{z}_{i,t}^s}_{\text{Stationary}} + 0$$
$$:= \epsilon_t^s.$$

- At future time step $t = t + 1, t + 2, \cdots, t + K$ (using world model to predict future steps observations),

$$\epsilon_t^{\text{Epistemic}} := \bar{z}_{i,t} - \hat{z}_{i,t}$$
$$\triangleq \underbrace{\bar{z}_{i,t}^s - \hat{z}_{i,t}^s}_{\text{Stationary}} + \underbrace{\bar{z}_{i,t}^{ns} - \hat{z}_{i,t}^{ns}}_{\text{Non-stationary}}$$
$$:= \epsilon_t^s + \epsilon_t^{ns}.$$

In what follows, we characterize the stationary and non-stationary part in the epistemic error, respectively.

**Stationary Part.** At each time step $t$, the agent will leverage the sequence model $f_\phi$ in the world model to generate imaginary trajectories $\{\hat{x}_{i,t+k}, a_{i,t+k}\}_{k=1}^K$ with prediction horizon $K > 0$, based on the model state $x_{i,t} := [z_{i,t}, h_{i,t}]$ and policy $a_{i,t} \sim \pi_i$. In our theoretical analysis, we consider the sequence model in the world model to be the RNN and follow the same line as in previous works

Lim et al. (2021); Wu et al. (2021). In particular,

$$h_{i,t+1} = Ax_{i,t} + \sigma_h(Wx_{i,t} + Ua_{i,t} + b)$$
$$z_{i,t+1} = \sigma_z(Vh_{i,t+1}), \tag{6}$$

where $\sigma_h$ is a $L_h$-Lipschitz element-wise activation function (e.g., ReLU Agarap (2018)) and $\sigma_z$ is the $L_z$-Lipschitz activation functions for the state representation. The matrices $A, W, U, V, b$ are trainable parameters.

Assume the gap between the (stationary part) state obtained by using LSI and the state using only local information to be a random variable $\epsilon_{x,t} := x_t - x_{i,t}$ with expectation $\mathcal{E}_{x,t}$, where $x_t$ is the model state obtained by using LSI. Then by using the Lipschitz properties of the activation functions and Assumption 1, we obtain the upper bound for the error in the stationary part as,

$$\epsilon_{i,t+k}^s \leq L_h L_z V(W\epsilon_{i,x,t+k} + UL_a\epsilon_{i,x,t+k-1}) + L_z VA\epsilon_{i,x,t+k}, \tag{7}$$
$$= (L_h L_z VW + L_z VA)\epsilon_{i,x,t+k} + L_h L_z VUL_a\epsilon_{i,x,t+k-1} \tag{8}$$

where $\epsilon_{i,x,t+k-1}$ is the state difference from last time step. Note that the second term on the LHS is due to the action error which is directly related to the state from previous time step.

**Non-stationary Part.** The prediction error of the future steps also result from the non-stationarity of the environment since the other agents may adapt their policies during the interaction. To this end, let the expected total-variation distance between the true state transition probability $P(z'|z, a)$ and the predicted one $\hat{P}(z'|z, a)$ be upper bounded by $\mathcal{E}_P$, i.e., $\mathbf{E}_\pi[D_{\mathrm{TV}}(P||\hat{P})] \leq \mathcal{E}_P$. Moreover, we denote the gap of the input model state at time step $t$ as a random variable $\epsilon_x$ with expectation $\mathcal{E}_x := \max_k \mathbf{E}[x_{t+k} - \hat{x}_{i,t+k}]$, where $x_t$ is the model state obtained by using LSI.

Following the same line as in Lemma B.2 Janner et al. (2019), we assume

$$\max_t E_{z \sim p^t(z)} D_{KL}(p(z' \mid z) \| \hat{p}(z' \mid z)) \leq \mathcal{E}_P,$$

and the initial distributions are the same.

Then we have the marginal state visitation probability is upper bounded by

$$\frac{1}{2}\sum_z |\rho^{t+k}(z) - \hat{\rho}^{t+k}(z)| \leq k\mathcal{E}_P.$$

Meanwhile, for simplicity, we define the following notations to characterize the prediction error due to the non-stationarity (such that the predicted MDP is different from the underlying real MDP).

$$\mathbf{E}[z_1^{t+k}] = \rho_1^k \geq 0,$$
$$\mathbf{E}[z_2^{t+k}] = \hat{\rho}_2^k \geq 0,$$
$$\epsilon_{t+k}^{ns} := z_1^{t+k} - z_2^{t+k},$$

where $\rho_1^k = \mathbf{E}_{z_1 \sim \rho^k(z_1)}[z_1] = \sum_z z\rho^k(z)$ is the mean value of the marginal visitation distribution at time step $t + k$ (starting from time step $t$).

Then we obtain the upper bound for the non-stationary part of the prediction error as follows,

$$\mathbf{E}[\epsilon_{t+k}^{ns}] := \rho_1^k - \hat{\rho}_2^k \leq 2kB_x\mathcal{E}_P$$

**Error Accumulation and Propagation.** We first recall the decomposition of the prediction error as follows.

$$\epsilon_{t-1} = \underbrace{\epsilon_{t-1}^s + \epsilon_{t-1}^{ns}}_{\text{Epistemic Error}} + \epsilon_{t-1}^{\mathrm{RNN}} \tag{9}$$

Bring Equation (8) to Equation (9) gives us (we omit the agent index $i$ for brevity),

$$\begin{aligned}
\epsilon_{t+k} &= \epsilon_{t+k}^{ns} + \epsilon_{t+k}^{\mathrm{RNN}} + \epsilon_{t+k}^s \\
&\leq \epsilon_{t+k}^{ns} + \epsilon_{t+k}^{\mathrm{RNN}} + L_h L_z V(W\epsilon_{x,t+k} + UL_a\epsilon_{x,t+k-1}) \\
&= \epsilon_{t+k}^{ns} + \epsilon_{t+k}^{\mathrm{RNN}} + L_h L_z VW\epsilon_{x,t+k} + L_h L_z VUL_a\epsilon_{x,t+k-1} \\
&= \epsilon_{t+k}^{ns} + \epsilon_{t+k}^{\mathrm{RNN}} + (L_h L_z VW + L_z VA)\epsilon_{i,x,t+k} + L_h L_z VUL_a\epsilon_{i,x,t+k-1} \\
&:= M_{t+k} + N\epsilon_{x,t+k-1} \tag{10}
\end{aligned}$$

where
$$M_{t+k} := \epsilon_{t+k}^{ns} + \epsilon_{t+k}^{RNN} + (L_h L_z VW + L_z VA)\epsilon_{x,t+k}$$
$$N := L_h L_z VU L_a.$$

Notice that in the stationary part of the prediction error, we have $|\epsilon_{x,t+k}| = |\epsilon_{t+k}|$. Furthermore, by abuse of notation, we apply Equation (10) recursively and obtain the relationship between the prediction error at $k$ rollout horizon and the gap from the input, i.e.,

$$\epsilon_{t+k} \leq M_{t+k} + N\epsilon_{t+k-1}$$
$$\leq M_{t+k} + NM_{t+k-2}$$
$$\leq \cdots$$
$$\leq \sum_{h=0}^{k} N^h M_{t+k-h} + N^k \epsilon_t$$

Taking expectation on both sides gives us,

$$\mathbf{E}[\epsilon_{t+k}] \leq \sum_{h=0}^{k} N^h \mathbf{E}[M_{t+k-h}] + N^k \mathbf{E}[\epsilon_t]$$
$$\leq \sum_{h=0}^{k} N^h (2hB_x \mathcal{E}_P + N_1 \mathcal{E}_x + \mathbf{E}[\epsilon_{t+h}^{RNN}]),$$

where $N_1 = L_h L_z VW + L_z VA$.

**The Upper Bound of the Prediction Error.** Then by invoking Markov inequality, we have the upper bound for $\epsilon_t$ with probability at least $1 - \delta$ as follows,

$$\epsilon_{t+k} \leq \sum_{h=1}^{k} N^h \left( \frac{1}{\delta}(2hB_x \mathcal{E}_P + N_1 \mathcal{E}_x) + \sqrt{\Psi_t(n, \delta)} \right) := \mathcal{E}_{\delta,t}$$

## C   PROOF OF PROPOSITION 1

**The Sub-optimality Gap Due to Prediction Error.** Given a policy $\pi$, we aim to quantify the gap of the value function between using underlying ground truth state (e.g., with LSI) and the predicted state. Assume the underlying true state is $s_0$ and the predicted state is $o_0$. Then we have the gap of the value function to be as follows,

$$v(s_0) - v(o_0) = \mathbf{E}_{a \sim \pi} \left[ \sum_{l=0}^{L-1} \gamma^l r(s_{t+l}, a_{t+l}, a'_{t+l}) + \gamma^L Q_{-1}(s_{t+L}, a_{t+L}, a'_{t+L}) \right]$$
$$- \mathbf{E}_{a \sim \pi} \left[ \sum_{l=0}^{L-1} \gamma^l r(o_{t+l}, a_{t+l}, a'_{t+l}) + \gamma^L Q_{-1}(o_{t+L}, a_{t+L}, a'_{t+L}) \right]$$
$$= \mathbf{E}_{a \sim \pi} \left[ \sum_{l=0}^{L-1} \gamma^l (r_{t+l} - \hat{r}_{t+l}) \right] + \gamma^L \mathbf{E}_{a \sim \pi} \left[ Q_{t+L} - \hat{Q}_{t+L} \right]$$

For simplicity, we denote the reward of the underlying real state as $r_{t+l} := r(s_{t+l}, a_{t+l}, a'_{t+l})$ and the reward based on prediction observation as $\hat{r}_{t+l} := r(o_{t+l}, a_{t+l}, a'_{t+l})$.

In the previous theorem, we have the bound for $\epsilon_t := s_t - o_t$. Now we will characterize the impact of $\epsilon_t$ on the sub-optimality gap. We first recall the following assumptions on the MDP considered in this work.

**Assumption 4** (MDP Regularity). *We assume that 1) The state space $\mathcal{X}$ is a compact subset of $\mathbb{R}^d$ and the action space is finite; 2) The one step reward $r(s, a)$ is $L_r$-Lipschitz, i.e., for all $s, s' \in \mathcal{S}$ and $a, a' \in \mathcal{A}$*

$$|r(s, a) - r(s', a')| \leq L_r(d_S(s, s') + d_A(a, a')),$$

where $d_S$ and $d_A$ is the corresponding metric in the state space and action space (both are metric space); 3) The policy $\pi$ is $L_\pi$-Lipschitz, i.e.,

$$d_A(\pi(\cdot|s) - \pi(\cdot|s')) \leq L_\pi d_S(s, s').$$

With Assumption Assumption 3 holds, we obtain the following bound,

$$r_l - \hat{r}_l \leq L_r(1 + L_\pi)\epsilon_{t+l}$$

$$Q_{t+L} - \hat{Q}_{t+L} \leq L_Q(1 + L_\pi)\epsilon_{t+L},$$

where $L_Q := \frac{L_r}{1-\gamma}$.

Then we have the upper bound for the sub-optimality gap as,

$$\mathbf{E}_a\left[Q(s_t, a_t) - \hat{Q}(o_t, a_t)\right] \leq \mathbf{E}_{a\sim\pi}\left[\sum_{l=0}^{L-1}\gamma^l L_r(1 + L_\pi)\epsilon_{t+l}\right] + \gamma^L \mathbf{E}_{a\sim\pi}\left[L_Q(1 + L_\pi)\epsilon_{t+L}\right]$$

$$= \sum_{l=0}^{L-1}\gamma^l L_r(1 + L_\pi)\mathbf{E}_{a\sim\pi}[\epsilon_{t+l}] + \gamma^L L_Q(1 + L_\pi)\mathbf{E}_{a\sim\pi}[\epsilon_{t+L}]$$

Additionally, we assume that $\max_t \mathcal{E}_{\delta,t} = \mathcal{E}_{\max}$, then we have the upper bound for the sub-optimality gap as follows,

$$\mathbf{E}_{a_t}[Q(s_t, a_t) - \hat{Q}(o_t, a_t)] \leq \left(\frac{1 - \gamma^{L-1}}{1-\gamma}L_r(1 + L_\pi) + \gamma^L L_Q(1 + L_\pi)\right)\mathcal{E}_{\max} := \epsilon$$

Let the right side equal to $\epsilon$, then we have that when the prediction error when using the information $T(I_t)$ satisfies the following condition, then we say the information $T(I_t)$ is Ego-centric $\epsilon$-approximate sufficient.

$$\mathcal{E}_{\max} \leq \frac{\epsilon}{\bar{M}}$$

$$\bar{M} := \frac{1 - \gamma^{L-1}}{1-\gamma}L_r(1 + L_\pi) + \gamma^L L_Q(1 + L_\pi)$$

### C.1 Consider the Function Approximation Error

The non-stationarity originates from the policies anticipated by the ego agent when estimating the future reward is different from the agent's real action taken after observing new information. In this regard, we revise the results in the multi-step lookahead planning literature Janner et al. (2019) and have the following upper bound of the prediction error related to the non-stationarity. We assume the resulting MDP to be $\hat{M}$ and the Total Variation distance bound is $\epsilon_P$, then we have, with probability at least $1 - \delta$ for $H$-step prediction, then the sub-optimality gap of the $Q$-function is upper bounded by,

$$|\hat{Q}_t - Q_t| \leq \frac{2}{(1 - \gamma^h)\delta}\left[C(\epsilon_P, H, \gamma) + \frac{\epsilon_v}{2} + \gamma^H \epsilon_v\right],$$

where $C(\epsilon_P, h, \gamma) = R_{\max}\sum_{t=0}^{h-1}\gamma^t t\epsilon_P + \gamma^h h\epsilon_P V_{\max}$.

## D Communication Bandwidth (in bytes) Requirements

We summarize the bytes requirements for various information available in the CARLA platform. In Figure 5, we show the bandwidth requirements during the testing. In average, it only requires 0.106 MB data transmission, which is significantly lower than the full observation case, which requires more than 5.417 MB data in a 230 vehicle system per 0.1 second.

**Hidden Variables from Memory Units** ($h$)

- Dimensions: 2048 (32-bit float tensors).
- Total size: $2048 \times 4$ bytes/dimension = $8,192$ bytes.

**Latent Variables from Encoder** ($z$)

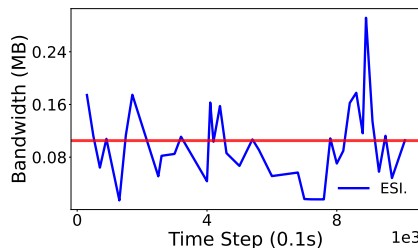

Figure 5: The bandwidth requirements (MB) of WM-MBRL in one testing. The red line is the average bandwidth requirements over all steps.

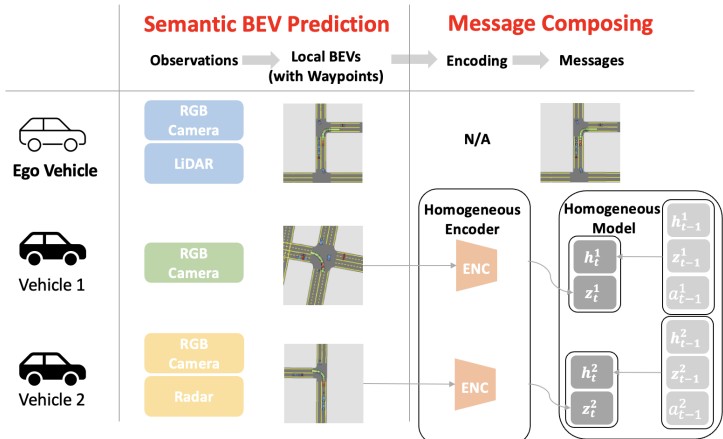

Figure 6: BEV encoding. In this case, each vehicle will generate the encoded messages based on its own BEV. The other agents are able to decode the model state directly by using their local decoder.

- Structure: 32 categorical variables with 32 classes each.

- Total size: $32 \times log_2(32) = 60$ bits = 7.5 bytes.

**Raw Image Time-Series Data**

- Assume 2K camera resolution: $2560 \times 1440$ pixels.

- For a $T$-step sequence: Size = $T \times 2560 \times 1440 \times 3$ bytes.

- This size is significantly larger than that of $h$ and $z$ variables.

**Model Weights or Gradients**

- Model parameters: Approximately 77 million.

- Total size: $77 \times 10^6$ parameters $\times 4$ bytes/parameter = $3.08 \times 10^8$ bytes.

**Bounding Boxes**

- Bounding boxes: four-tuple $(x, y, \text{height}, \text{width})$, each an int32.

- For $n$ bounding boxes: Total size = $4 \times 4$ bytes $\times n$ boxes = $16n$ bytes.

**Waypoints** Each waypoint is represented as a relative coordinate in the 128x128 BEV, and it will take approximately 14 bits.

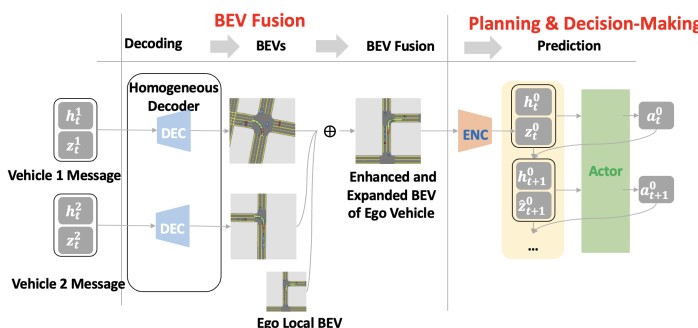

Figure 7: BEV decoding. In the setting where agents share the same encoder-decoder in the WM, the agents are able to decode the state from other agents directly using the locol decoder.

# E    EXPERIMENTS DETAILS

**Open Source.** Each agent is trained on one Nvidia A100 GPU. The overall training time for the agent is about 20 hours. The source code and numerical results will be open sourced. The overall flowchart of the proposed *CALL* is depicted in Figure 6 and Figure 7.

**Demo videos and images.** We provide the detailed demo videos and images in the supplementary materials.

**Report of Standard Deviation in Figures.** Note that all the learning curves presented in this work are smoothed by using exponential moving average with smoothing factor 0.72, with the shaded area to be the standard deviation. We use the smoothing algorithm provided by wandb.ai platform.

## E.1    TRAINING ENVIRONMENT AND TASKS.

**Local Trajectory Planning.** The conventional approach to studying autonomous driving involves distinct modules, comprising perception, planning, and control. Perception aims to extract relevant information for autonomous vehicles from their surroundings. Control is responsible for determining optimal actions such as steering, throttle, or brake, ensuring the autonomous vehicles adhere to the planned path. The primary objective of planning is to furnish vehicles with a secure and collision-free path toward their destinations, considering vehicle dynamics, maneuvering capabilities in the presence of obstacles, and adherence to traffic rules and road boundaries. In our work, we take an end-to-end approach to address the local trajectory planning tasks for autonomous driving and focus on the lower-level of decision making while adhering to the planned waypoints. The goal of the vehicle is to generate a sequence of actions in order to travel along the planned waypoints while following the travel rules (e.g., speed limit) and avoiding collision with other vehicles. In particular, we consider the decision making problem in the multi-agent system as a stochastic game with state space, action space and reward defined as follows.

**State Space.** There are two parts of information considered as vehicle's state. First, the environment observation from sensors such as cameras, radar and LiDAR, which captures the objects in the environments and corresponding geographical information. Meanwhile, the other vehicles behavior information, such as their waypoins as action intention. Following the standard approach Bansal et al. (2018); Chen et al. (2023), we use a BEV semantic segmentation image with size of 128 as the unified state representation of the state. For instance, in Figure 8, the blue line represents ego vehicle's planned waypoints. The yellow line is the other vehicle's planned waypoints. The ego vehicle is marked in red while the other vehicle that can be observed by ego agent is marked in green. The vehicle in yellow represents the blocked vehicle (from ego agent's perspective) but can be 'observed' by using shared information. The gray vehicle is not visible for ego agent.

**Action Space.**    In our experiments, we consider the discrete action space.    Particularly, at each time step, the agent needs to choose acceleration and steering angle from $[-2, 0, 2]$ and $[-0.6, -0.2, 0, 0.2, 0.6]$, respectively.

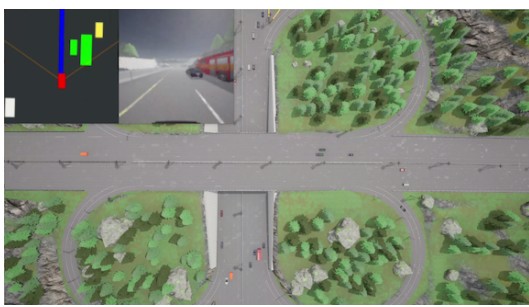

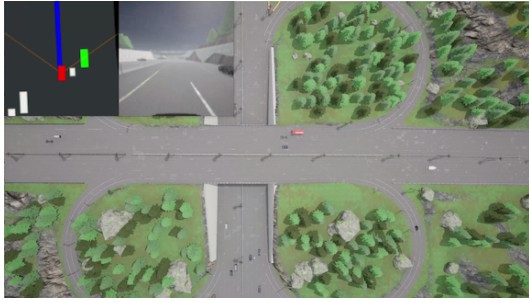

Figure 8: Examples of BEV representations (top left square).

**Reward.** We design the reward as the weighted sum of six different factors, i.e.,

$$R_t = w_1 R_{\text{safe}} + w_2 R_{\text{comfort}} + w_3 R_{\text{time}} + w_4 R_{\text{velocity}} + w_5 R_{\text{ori}} + w_6 R_{\text{target}},$$

In particular,

- $R_{\text{safe}}$ is the time to collision to ensure safety

- $R_{\text{comfort}}$ is relevant to jerk behavior and acceleration

- $R_{\text{time}}$ is to punish the time spent before arriving at the destination

- $R_{\text{velocity}}$ is to penalize speeding when the velocity is beyond 5m/s and the leading vehicle is too close

- $R_{\text{ori}}$ is to penalize the large orientation of the vehicle

- $R_{\text{target}}$ is to encourage the vehicle to follow the planned waypoints

**Number of Vehicles.** In our experiments, we consider the number of vehicles to be 150 and 250, respectively, to demonstrate the scalability of the proposed *CALL* framework. The agents determine their waypoints for the next steps while updating them during the interaction. All the experiments are conducted in CARLA Town04 as shown in Figure 8. In this section, we summarize the experiment results in the 250 vehicles systems.

### E.2 WORLD MODEL TRAINING

We use Dreamer v3 Hafner et al. (2023) structure, i.e., encoder-decoder, RSSM Hafner et al. (2019), to train the world model and adopt the large model for all experiments with dimension summarized in Table 4. We first restate the hyper-parameters in Table 5.

**Learning BEV Representation.** The BEV representation can be learnt by using algorithms such as BevFusion Liu et al. (2023), which is capable of unifying the cameras, LiDAR, Radar data into a BEV representation space. In our experiment, we leverage the privileged information provided by CARLA Dosovitskiy et al. (2017), such as location information and map topology to construct the BEVs.

| Dimension | L |
|---|---|
| GRU recurrent units | 2048 |
| CNN multiplier | 64 |
| Dense hidden units | 768 |
| MLP layers | 4 |
| Parameters | 77M |

Table 4: Model Sizes Hafner et al. (2023).

| Name | Symbol | Value |
|---|---|---|
| **General** | | |
| Replay capacity (FIFO) | — | $10^6$ |
| Batch size | $B$ | 16 |
| Batch length | $T$ | 64 |
| Activation | — | $\text{LayerNorm} + \text{SiLU}$ |
| **World Model** | | |
| Number of latents | — | 32 |
| Classes per latent | — | 32 |
| Reconstruction loss scale | $\beta_{\text{pred}}$ | 1.0 |
| Dynamics loss scale | $\beta_{\text{dyn}}$ | 0.5 |
| Representation loss scale | $\beta_{\text{rep}}$ | 0.1 |
| Learning rate | — | $10^{-4}$ |
| Adam epsilon | $\epsilon$ | $10^{-8}$ |
| Gradient clipping | — | 1000 |
| **Actor Critic** | | |
| Imagination horizon | $H$ | 15 |
| Discount horizon | $1/(1-\gamma)$ | 333 |
| Return lambda | $\lambda$ | 0.95 |
| Critic EMA decay | — | 0.98 |
| Critic EMA regularizer | — | 1 |
| Return normalization scale | $S$ | $\text{Per}(R, 95) - \text{Per}(R, 5)$ |
| Return normalization limit | $L$ | 1 |
| Return normalization decay | — | 0.99 |
| Actor entropy scale | $\eta$ | $\mathbf{3 \cdot 10^{-4}}$ |
| Learning rate | — | $3 \cdot 10^{-5}$ |
| Adam epsilon | $\epsilon$ | $10^{-5}$ |
| Gradient clipping | — | 100 |

Table 5: Dreamer v3 hyper parameters Hafner et al. (2023).

**World Model Training.** The world model is implemented as a Recurrent State-Space Model (RSSM) Hafner et al. (2019; 2023) to learn the environment dynamics, encoder, reward, continuity and encoder-decoder. We list the equations from the RSSM mode as follows:

$$
\text{RSSM} \begin{cases}
\text{Sequence model:} & h_t = f_\phi(h_{t-1}, z_{t-1}, a_{t-1}) \\
\text{Encoder:} & z_t \sim q_\phi(z_t | h_t, x_t) \\
\text{Dynamics predictor:} & \hat{z}_t \sim p_\phi(\hat{z}_t | h_t)
\end{cases}
$$

$$
\begin{aligned}
\text{Reward predictor:} \quad & \hat{r}_t \sim p_\phi(\hat{r}_t | h_t, z_t) \\
\text{Continue predictor:} \quad & \hat{c}_t \sim p_\phi(\hat{c}_t | h_t, z_t) \\
\text{Decoder:} \quad & \hat{x}_t \sim p_\phi(\hat{x}_t | h_t, z_t)
\end{aligned}
\tag{11}
$$

We follow the same line as in Dreamer v3 Hafner et al. (2023) to train the parameter $\phi$. We include the following verbatim copy of the loss function considered in their work.

Given a sequence batch of inputs $x_{1:T}$, actions $a_{1:T}$, rewards $r_{1:T}$, and continuation flags $c_{1:T}$, the world model parameters $\phi$ are optimized end-to-end to minimize the prediction loss $\mathcal{L}_{\text{pred}}$, the dynamics loss $\mathcal{L}_{\text{dyn}}$, and the representation loss $\mathcal{L}_{\text{rep}}$ with corresponding loss weights $\beta_{\text{pred}} = 1$, $\beta_{\text{dyn}} = 0.5$, $\beta_{\text{rep}} = 0.1$:

$$\mathcal{L}(\phi) \doteq \mathbf{E}_{q_\phi}\left[\sum_{t=1}^{T}(\beta_{\text{pred}}\mathcal{L}_{\text{pred}}(\phi) + \beta_{\text{dyn}}\mathcal{L}_{\text{dyn}}(\phi) + \beta_{\text{rep}}\mathcal{L}_{\text{rep}}(\phi))\right]. \tag{12}$$

$$\mathcal{L}_{\text{pred}}(\phi) \doteq -\ln p_\phi(x_t|z_t, h_t) - \ln p_\phi(r_t|z_t, h_t) - \ln p_\phi(c_t|z_t, h_t)$$

$$\mathcal{L}_{\text{dyn}}(\phi) \doteq \max\big(1, \text{KL}[\text{sg}(q_\phi(z_t|h_t, x_t))|| \quad p_\phi(z_t|h_t)]\big) \tag{13}$$

$$\mathcal{L}_{\text{rep}}(\phi) \doteq \max\big(1, \text{KL}[\quad q_\phi(z_t|h_t, x_t) ||\text{sg}(p_\phi(z_t|h_t))]\big)$$

**Actor-Critic Learning.** We consider the prediction horizon to be 16 as the same as in Dreamer v3 while training the actor-critic networks. We follow the same line as in Dreamer v3 and consider the actor and critic defined as follows.

$$\begin{aligned} \text{Actor:} &\quad a_t \sim \pi_\theta(a_t|x_t) \\ \text{Critic:} &\quad v_\psi(x_t) \approx \mathbf{E}_{p_\phi, \pi_\theta}[R_t], \end{aligned} \tag{14}$$

where $R_t \doteq \sum_{\tau=0}^{\infty} \gamma^\tau r_{t+\tau}$ with discounting factor $\gamma = 0.997$. Meanwhile, to estimate returns that consider rewards beyond the prediction horizon, we compute bootstrapped $\lambda$-returns that integrate the predicted rewards and values:

$$R_t^\lambda \doteq r_t + \gamma c_t\Big((1-\lambda)v_\psi(s_{t+1}) + \lambda R_{t+1}^\lambda\Big) \qquad R_T^\lambda \doteq v_\psi(s_T) \tag{15}$$

### E.3   CHOICE OF DATASET AND BASELINE

**Choice of Baseline.** Our choice of baselines was guided by several important considerations:

- First, we focused on world model-based approaches specifically designed for autonomous driving tasks, given the unique challenges of the high-dimensional CARLA environment. Many conventional RL approaches struggle with the curse of dimensionality in such settings without substantial modifications. We choose the SOTA work just published in 2024 Li et al. (2024) on autonomous driving planning, which is based on DreamerV3, as our primary baseline (denoted as 'Local Obs.' in Figure 3(a)). Additionally, we included a variant without waypoint sharing (LSI) for ablation studies of the impact of our communication mechanism.

- The works in Table 1 either not using world model (hence not being able to effectively deal with high-dimensional inputs in CARLA), or lack of intention sharing (which is essential for planning) or have requirements on for sharing all information among agents (hence impractical for a large multi-agent systems as considered in our work). While the works by Pan et al. (2022); Liu et al. (2024) are world model based methods, they were developed for fundamentally different environments, i.e., the DeepMind Control Suite and SMAC benchmark respectively. Adapting these methods to CARLA's autonomous driving setting would require significant architectural modifications that could compromise their original design principles. For instance, both [R1,R2] do not have dedicated module for intention process, which is critical for autonomous driving to understand the potential actions of other agents in the environment.

- To ensure fair comparison, we believe it's more appropriate to compare against methods specifically designed for similar autonomous driving scenarios, and in this case, Think2drve (Dreamerv3 based) approach is the SOTA on solving planning in CARLA benchmart. To our knowledge, CALL represents the first multi-agent world model-based approach specifically designed for autonomous driving tasks.

**Choice of Benchmark.** CARLA presents substantially more challenging scenarios compared to traditional multi-agent benchmarks like DeepMind Control Suite and SMAC, particularly due to its realistic vehicle dynamics and multi-agent interactions that follow traffic rules and safety protocols. Meanwhile, the planning in CARLA generally need longer-horizon and prediction (3-5 seconds ahead) versus shorter planning horizons as in other benchmarks.

While CALL's core principles of distributed learning, prediction-driven communication, and ego-centric world models, are indeed applicable to other multi-agent scenarios, we chose autonomous

driving as our primary test case due to its compelling combination of real-world significance and rigorous requirements for safety, efficiency, and scalability. The successful demonstration of CALL in this challenging environment provides strong evidence for its potential effectiveness in other multi-agent settings.

### E.4    SUPPLEMENTARY EXPERIMENT RESULTS

**Pre-training.** We warm-start our agents to facilitate the training speed. The pre-trained model is obtained from trajectory planning tasks with 50 background vehicles and a fixed ego path. The BEVs consist of all the vehicles without waypoints and are used as inputs. We migrate the model after 80k steps to more a complex setting with 150 vehicles, and 170k steps to the setting with 250 vehicles and random ego paths.

**Larger Scale Experiments.** To validate the scalability of the proposed *CALL*, we consider the challenging setting in the CARLA simulator with 250 agents. The learning performance and ablation studies are summarized below.

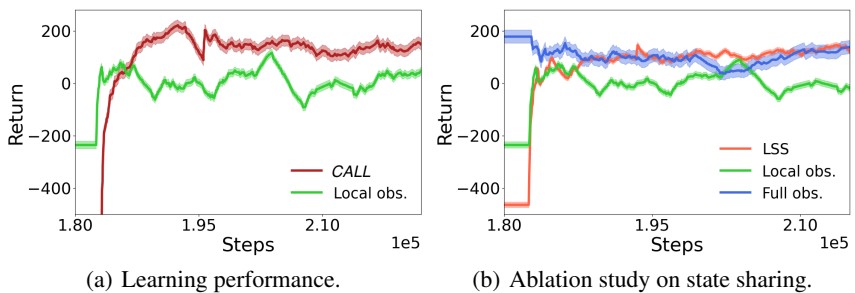

(a) Learning performance.          (b) Ablation study on state sharing.

Figure 9: The learning performance comparison and the ablation study on the model state.

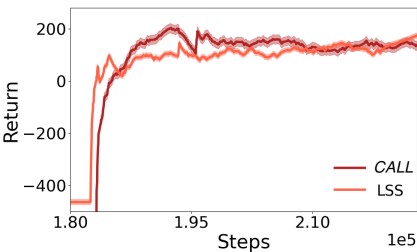

Figure 10: The ablation studies on the waypoints sharing.

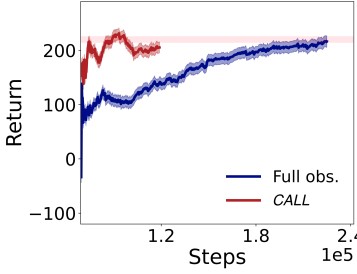

Figure 11: Learning Speed Comparison. It can be seen that at around 230k, the full observation with WP setting reaches the same return as in LSI setting. We include the standard error in the figures (shaded area) coming from the exponential moving average smoothing process with parameter 0.72 (we use the same smoothing code as the one provided by wandb.ai platform).

**Evaluation Metrics** We evaluate the performance of the ego agent in the multi-agent system, where we assume the RL agents in the system have the same world model, e.g., encoder-decoder and RSSM Hafner et al. (2023). Each evaluation session contains 15k steps. In particular, we consider the following metrics. The testing results are summarized in Table 6.

- **Percentage of successes**: the percentage of the waypoints that the car successfully reached.
- **Average TTC**: the average Time to Collision during the testing episode
- **Collision Rate**: The percentage of collision steps over all the evaluation steps.

| Metric | Success Rate | Average TTC | Collision Rate |
|---|---|---|---|
| Full Observation | 80% | 2.233 | 0.59 % |
| LSI | 87% | 2.845 | 0.28 % |
| LSS | 52% | 1.52 | 0.63 % |

Table 6: Testing Results.

**Evaluation Curve.** We summarize the agent's performance in the same testing environment with different settings in Figure 12. It can be seen that LSI and full observation setting reach the very similar return during the evaluation, while both are better than local information setting.

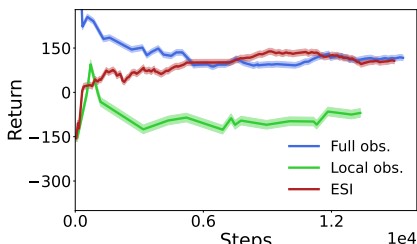

Figure 12: Evaluation curves in three settings: Local observation, Full observation and *CALL*.

### E.5 WORLD MODEL'S GENERALIZATION CAPABILITY

In Figures 13 and 14, we provide more examples of the prediction results during training stage using the world model.

**World Model's Generalization Capability in the Seen Environment with Changing Background Traffics.** We train the world model within a four-lane road section in CARLA Town04. The total distance between the source and destination endpoints is around 150m. To evaluate the generalization capability of the world model, we randomly generate source and destination endpoints, lane changing points, and background traffic (ref. Figure 15).

**World Model's Generalization Capability in the Unseen Environment.** Next, we evaluate the world model's generalization capability in unseen road sections such as the two-lane section and crossroad. The evaluation results in Figure 16 show that the world model can generalize to various environments without compromising the overall performance.

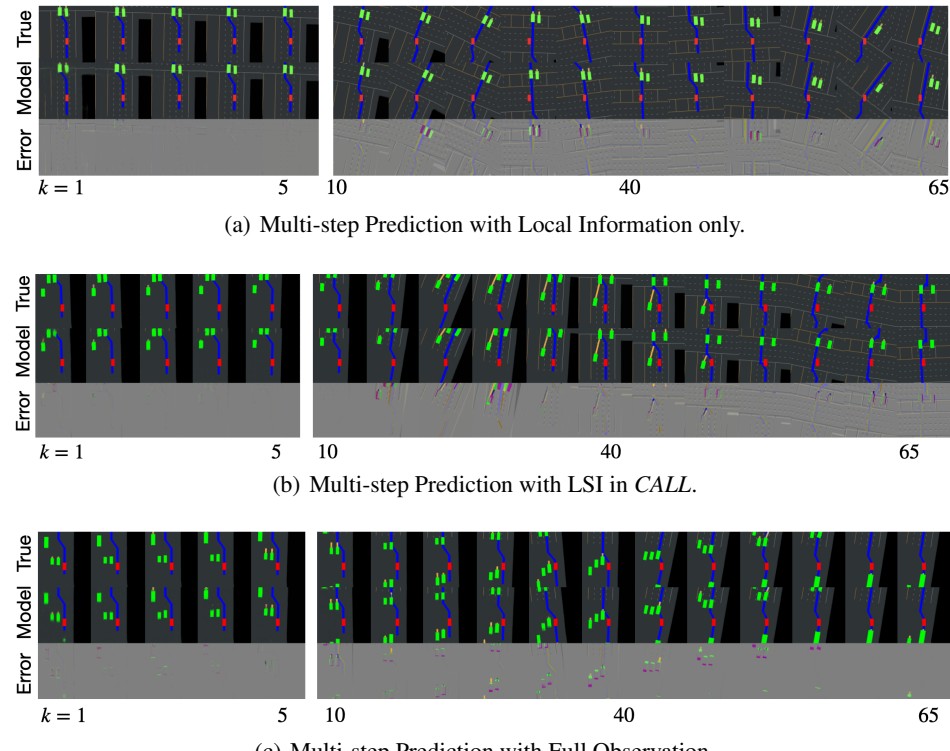

(a) Multi-step Prediction with Local Information only.

(b) Multi-step Prediction with LSI in *CALL*.

(c) Multi-step Prediction with Full Observation.

Figure 13: The comparison of the BEV multi-step prediction results with different information settings.

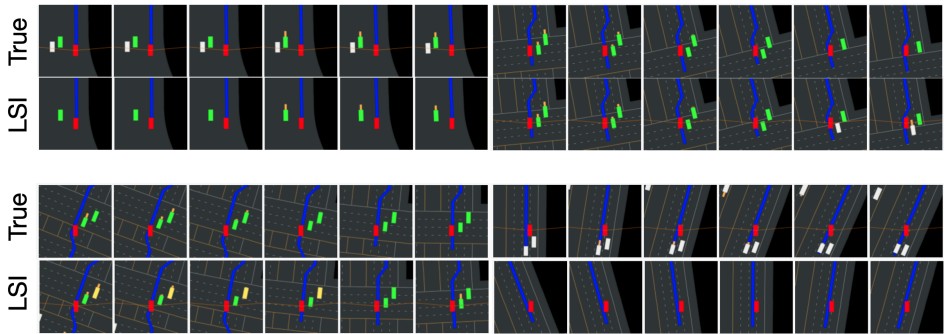

Figure 14: Comparison of underlying true BEV and LSI BEV.

## F    THE HETEROGENEOUS CASE: DIFFERENT AGENTS HAVE DIFFERENT WORLD MODELS

In this case, the WMs vary across agents and, therefore, latent spaces may be different. As a result, the shared latent representation is not decodable. To resolve this issue, it is plausible for each agent to first map local high-dimensional sensory inputs to semantic BEVs, in a cross-modal manner. Since semantic BEVs are interpretable by all vehicle agents (BEV can be viewed as a common language by vehicles), agents of interest can share local BEVs, which can then be fused, together with waypoints, into an enhanced and expanded BEV for the ego agent. As illustrated in Figure 17, the fused BEV can be then encoded into latent representation by WM to improve prediction and planning.

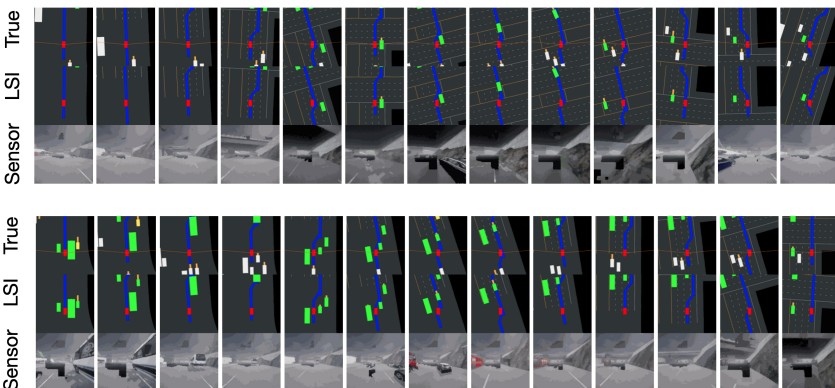

Figure 15: Evaluation of World Model's Generalization Capability in the four-lane road section with randomly generated traffic and ego paths.

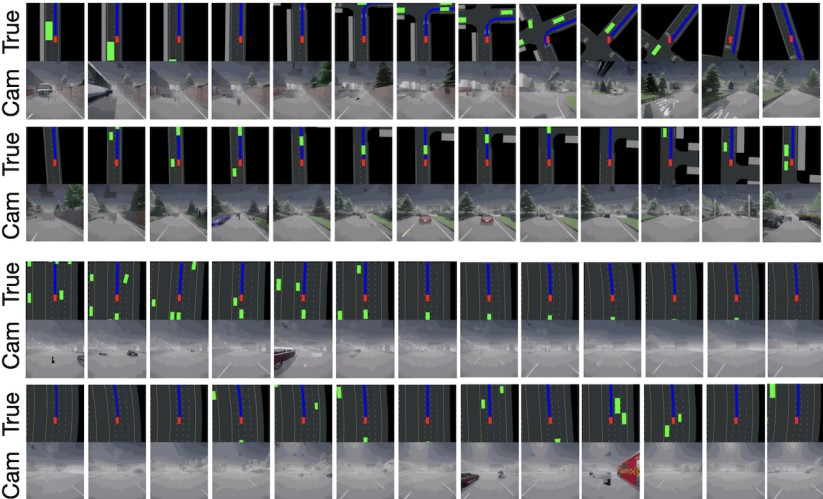

Figure 16: Evaluation of World Model's Generalization Capability in the unseen environment.

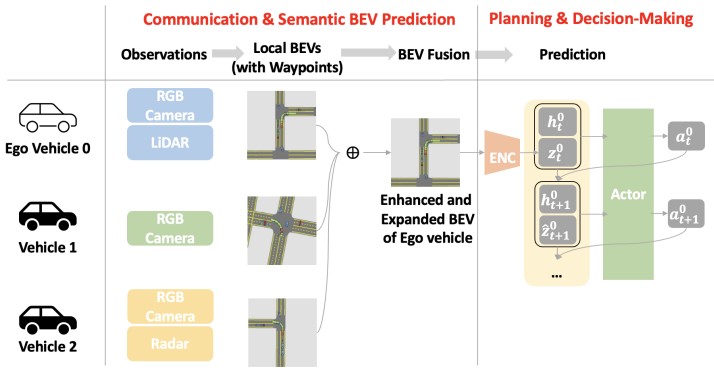

Figure 17: Heterogeneous World Model Setting. In this setting, agents are equiped with different encoder-decoders.

# G  IMPACT OF THE PREDICTION ACCURACY THRESHOLD $c$

We first summarize the prediction accuracy driven mechanism as follows:

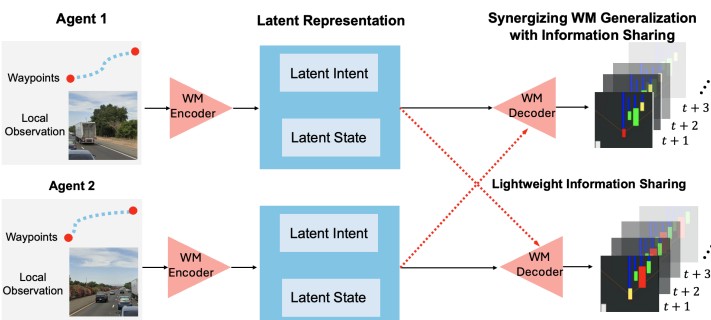

Figure 18: An illustration of *CALL* in a two-agent case: Each agent encodes high-dimensional sensory inputs and planned waypoints into low-dimensional latent state and latent intent, which can be shared via lightweight communications (e.g., red dashed arrow) and used as inputs to enrich perception and planning. Aided by information sharing, the generalization capabilities of world models lend each agent the power of foresight, enabling it to obtain better prediction of future environment dynamics in multi-agent systems.

- Step 1: Each agent continuously monitors its prediction performance by comparing predicted latent states and intentions against actual observations over the past $K$ time-steps.

- Step 2: When prediction errors exceed a threshold $c$, the agent automatically increase its communication range by 5 meters and initiates selective information exchange with relevant neighboring agents. Otherwise, the agent will remain its current communication range for information exchange.

Our empirical analysis demonstrates the critical relationship between the prediction accuracy threshold $c$ and system performance. Figure 19(a) reveals that at 120k training steps, very low $c$ values necessitate near-complete network communication, approaching centralized implementation with substantial bandwidth requirements ( 5MB). However, this extensive information sharing does not translate to optimal performance, likely due to the inclusion of non-essential or potentially noisy information that may impede efficient learning. We observe that performance generally improves as $c$ increases from 0 to 50, reaching peak efficiency in the range $c \in [10, 80]$, before declining for larger values. At the extreme ($c \to \infty$), agents operate in isolation without communication, leaving the fundamental challenges of partial observability and non-stationarity in MARL unaddressed.

Figure 19(b) illustrates the relationship between communication bandwidth and the prediction accuracy threshold. Higher $c$ values indicate greater tolerance for prediction errors, resulting in more selective information sharing. Notably, when $c = 50$, the communication bandwidth requirements are approximately 50 times lower than the full observation case, while maintaining strong performance. This demonstrates that CALL achieves efficient communication without sacrificing effectiveness. Furthermore, the broad range of c values yielding good performance ($c \in [10, 80]$) suggests that the algorithm is robust to threshold selection, making it practical for real-world implementation.

## H    ILLUSTRATIONS OF PREDICTION ERROR

Next, we present a comparative analysis of prediction errors across three different methods Figures 20 and 21: CALL, Local Observation Only, and Full Observation. The prediction error is calculated by comparing the pixels difference between the predicted BEV and the underlying true BEV obtained at the later steps. In particular, we evaluate the prediction error as follows,

- For a single pixel $(i, j)$: $E(i, j) = |P(i, j) - G(i, j)|$

- For the entire image with size $H \times W$: $E_{total} = \frac{1}{H \times W} \sum_{i=1}^{H} \sum_{j=1}^{W} |P(i, j) - G(i, j)|$

Our experiments demonstrate distinct error patterns over 30 steps, revealing several key insights. In both single-step and accumulated error evaluation, the Local Observation Only method consistently shows higher prediction errors, particularly in the later steps where it reaches peaks of approximately 250 and 290 units respectively. The CALL method and Full Observation approach exhibit more

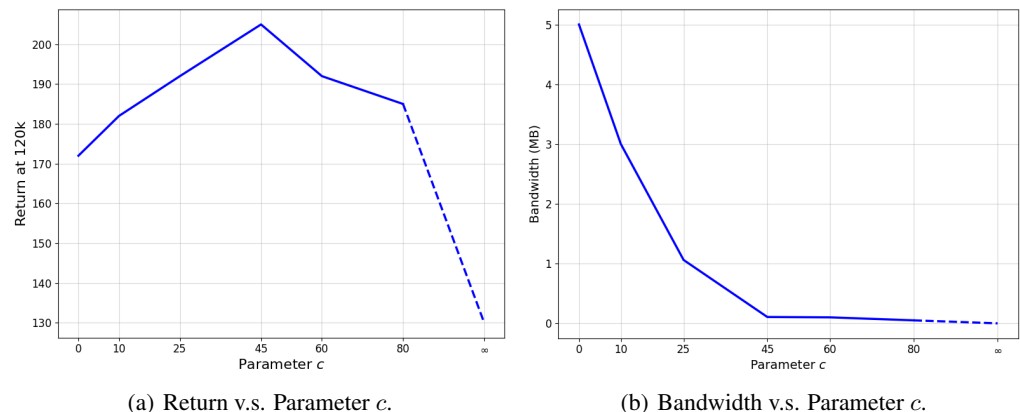

(a) Return v.s. Parameter $c$.     (b) Bandwidth v.s. Parameter $c$.

Figure 19: The impact of parameter $c$ on average return and the bandwidth requirements for communication.

stable error patterns, with CALL typically maintaining intermediate error levels between the other two methods. In Figures 20(b) and 21(b), we consider the 250 agents case. Our results show larger magnitudes of errors across all methods compared to the 150 agents case, indicating that the prediction task in the second scenario was more challenging. This pattern is particularly evident in the Local Observation Only method, which shows more pronounced error increases after step 20 in the second experiment.

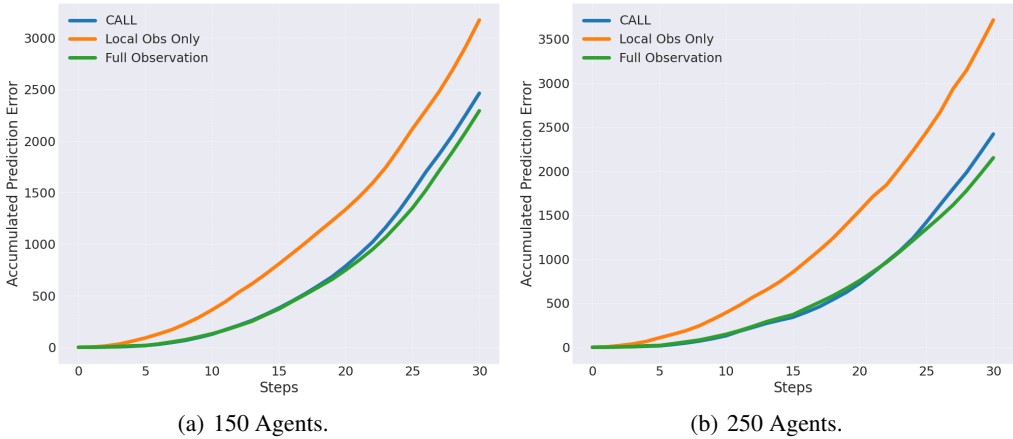

(a) 150 Agents.     (b) 250 Agents.

Figure 20: The comparison of accumulation error in the 30 steps predictions.

# I   ILLUSTRATION OF PREDICTION-ACCURACY GUIDED COMMUNICATION IN CALL

The CALL algorithm demonstrates effective adaptive communication through its prediction-error guided approach, as shown in the comparison between local observation, full observation, and CALL variants in Figures 22 and 23 (the prediction error curves are smoothed by a window size 5). In the case with 150 agents, the local observation method shows initially high prediction errors around 75-80, which gradually decreases but remains volatile throughout the training process. In contrast, the full observation approach, while more stable, maintains a relatively high prediction error averaging around 40-50. The CALL algorithm achieves a balance between these extremes by adaptively adjusting its communication range when prediction errors exceed 45, evaluated every 2k

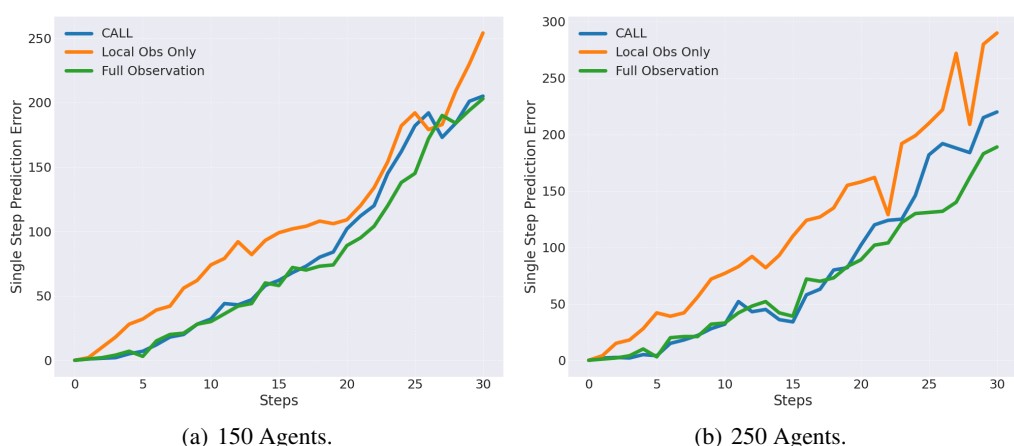

(a) 150 Agents.

(b) 250 Agents.

Figure 21: The comparison of single-step prediction errors.

steps for stability. This results in a more efficient communication strategy that maintains prediction accuracy while reducing unnecessary information sharing.

The performance differences become more pronounced in the larger scale scenario with 250 agents, where the benefits of CALL's adaptive communication become more apparent. The local observation approach continues to show high volatility and prediction errors, while the full observation method, despite having access to complete information, doesn't necessarily translate to better performance due to the increased complexity of processing more agent information. The CALL algorithm maintains its efficiency by strategically updating its communication range, starting from a small range and incrementally adjusting it based on prediction error thresholds. This adaptive approach helps CALL achieve lower prediction errors, particularly in the later stages of training (beyond 100k steps), while maintaining a more stable performance compared to both baseline approaches.

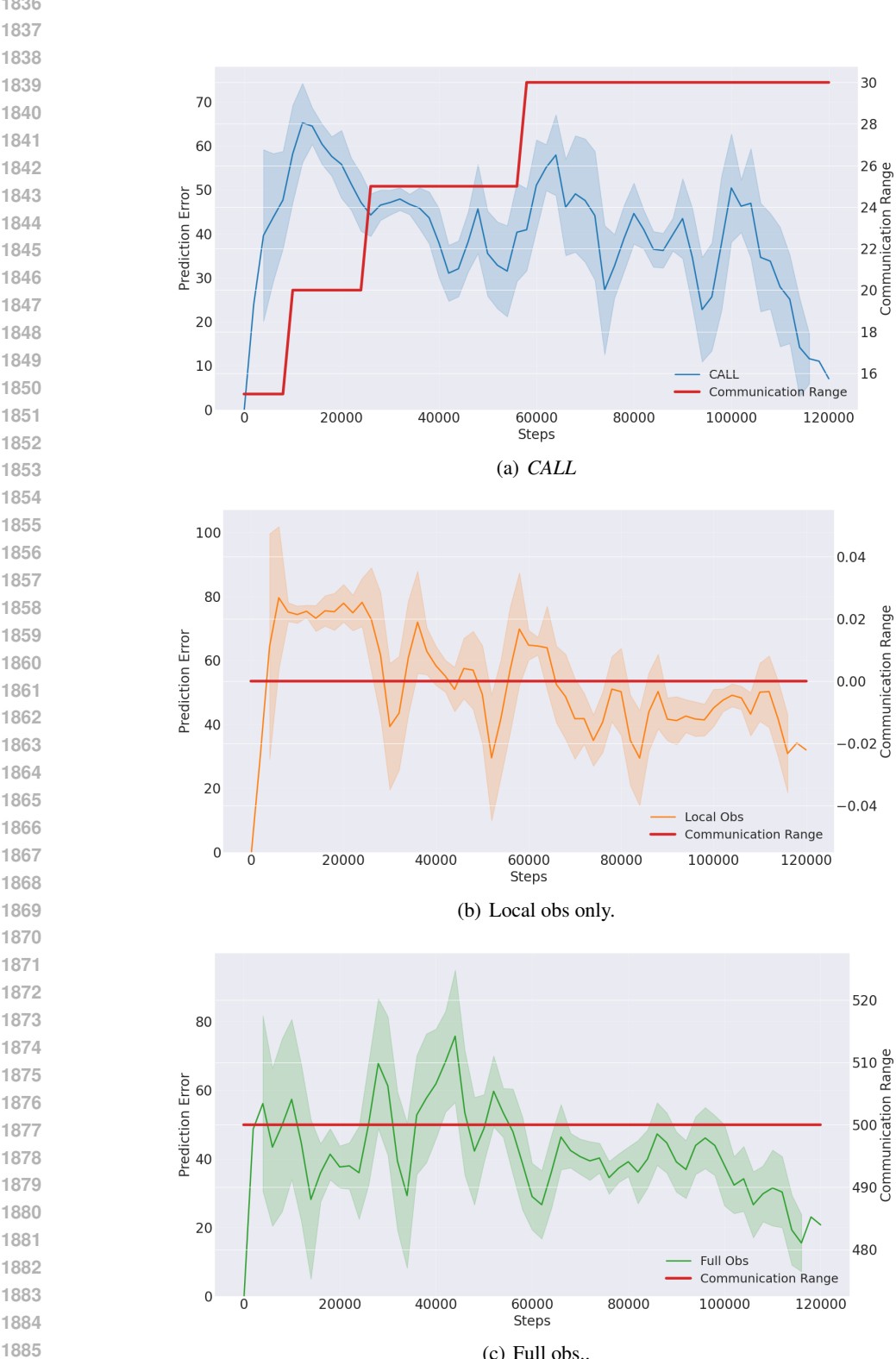

(a) *CALL*

(b) Local obs only.

(c) Full obs..

Figure 22: The comparison of prediction errors and communication ranges in 150 agents case.

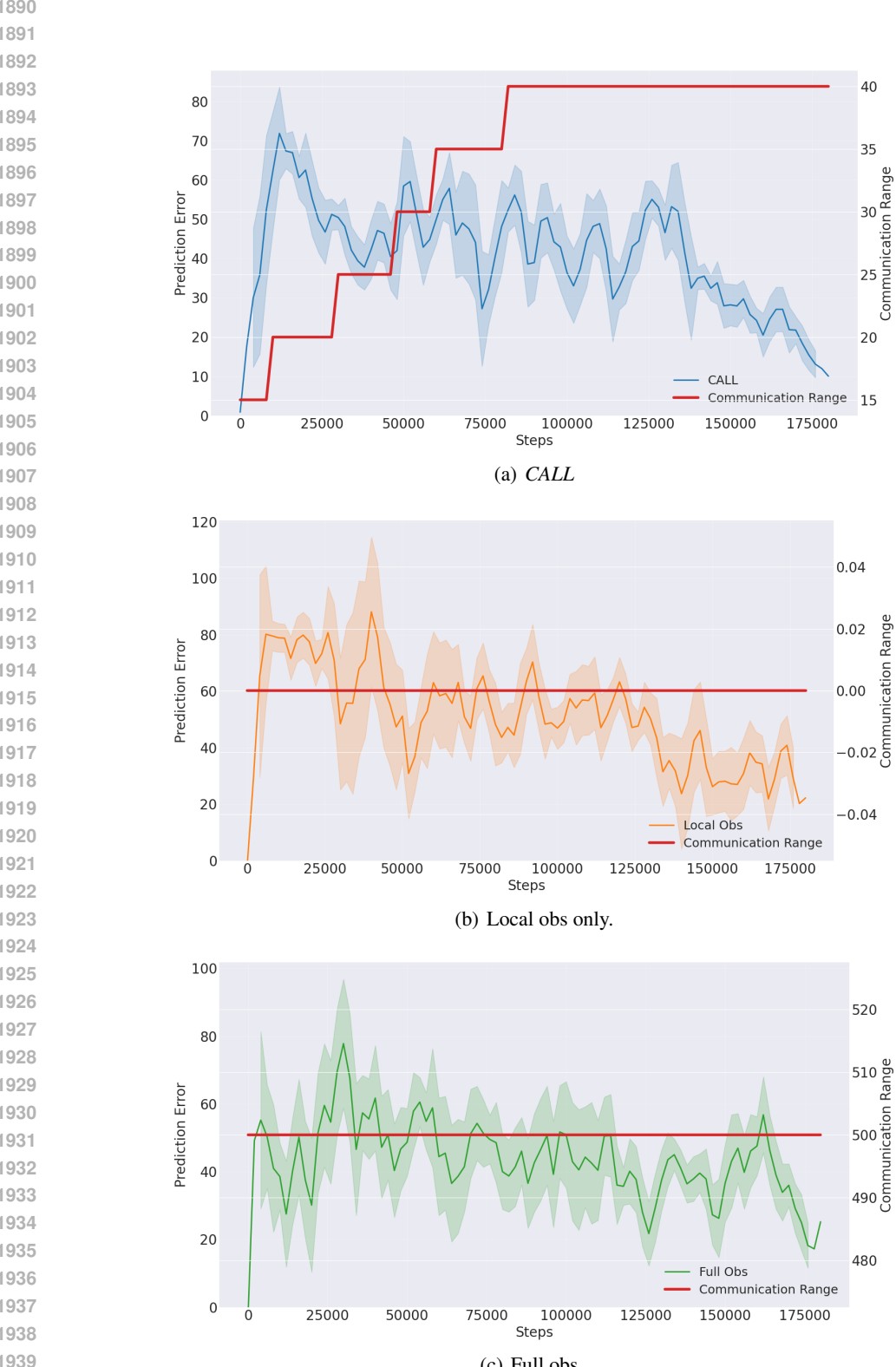

(a) *CALL*

(b) Local obs only.

(c) Full obs..

Figure 23: The comparison of prediction errors and communication ranges in 250 agents case.

