# OpenReview forum: "Ego-centric Learning of Communicative World Models for  Autonomous Driving"
_ICLR.cc/2025/Conference — Submitted to ICLR 2025_

### Official Review · Reviewer_nMor · 2024-10-27

**Soundness:** 2
**Presentation:** 3
**Contribution:** 2
**Rating:** 3
**Confidence:** 4

**Summary:**

This paper developed CALL, Communicative World Model, for ego-centric MARL. The CALL allows agents to adaptively adjust their information sharing based on real-time evaluation of their prediction accuracy, and reduce unnecessary information transmission and improve system efficiency.

**Strengths:**

**Pros:**

1. Theoretical Support

The article provides theoretical analysis proving the impact of prediction errors on sub-optimality gaps and demonstrates the effectiveness of the CALL method through experiments.

2. Experimental Validation

Extensive experiments were conducted on the CARLA platform for trajectory planning tasks, showing that the use of CALL can significantly improve performance, particularly in lightweight communication scenarios.

**Weaknesses:**

**Cons:**

1. Inaccurate Description of Literature

The article states: “Recent works on WM-based reinforcement learning still face significant limitations, and often rely on rigid, static information-sharing mechanisms, such as sharing information with all agents (Pretorius et al., 2020), using centralized frameworks (Krupnik et al., 2020; Pan et al., 2022; Liu et al., 2024).” In fact, Pan et al. (2022) and Liu et al. (2024), among others, learned and planned in the latent space. Therefore, lightweight information sharing is not unique to this paper, as most related works do this.

2.  Questions about Scalability and Adaptability

 The method described in the article establishes a world model for each agent. Since it sets the parameters and number of models beforehand, the method cannot adapt to changes in the number of agents during training or testing, especially during testing. Additionally, changes in the number of agents also change the observation dimensions for each agent, which the method cannot adapt to dynamically (dynamic changes in relevant information). When scaled up to large systems, the increase in the number of models and parameters with the number of agents leads to increased computational complexity and resource burden, thus limiting scalability. Although an experiment with 250 vehicles was conducted in Section E.3, there is no comparative analysis with the results from the experiment involving 150 vehicles, making it difficult to prove its good scalability.

3. Latent Intention

 There is no description of the definition and learning method of latent intention throughout the paper. Moreover, equations 11-13 also do not have terms related to informations T and latent intentions w.  Therefore, how to infer and learn these latent intentions?

 4. Comparative Experiments

  There is a lack of comparative experiments with related methods, such as those by Pretorius et al. (2020), Krupnik et al. (2020), Pan et al. (2022), and Liu et al. (2024).

 5. Experimental Verification of Accumulative Error in Multi-step Prediction

 Although the method theoretically proves its effectiveness, there is a lack of direct experimental verification. Specifically, there are no curves showing the relationship between the number of prediction steps and cumulative error, comparison curves of cumulative error with other methods, or performance change curves at different prediction step numbers.

6.  Prediction Accuracy-Driven Information Sharing

The article claims that CALL allows agents to adaptively adjust their information sharing based on real-time evaluation of their prediction accuracy. However, the specific mechanism of how the method adjusts information sharing according to prediction accuracy has not been experimentally verified, i.e., there are no graphs showing the relationship between prediction accuracy and information sharing (T).

**Questions:**

See the weakness  for Questions.

**Suggestions for Improvement:**

1. Improve the accuracy of literature description

2. Define intention clearly and explain learning mechanism

3. Add comparative experiments.

4. Supply experiments concerning accumulative errors in multi-step prediction and relationship between prediction accuracy and information (T)

---

> ### Author Response · Authors · 2024-11-19
> **Reply to Reviewer nMor (1/2)**
>
> We thank the reviewer nMor for your review and feedback. We address the concerns raised as follows.
>
> ### Weakness
>  **A1. Description of Literature**
>
> We appreciate the reviewer's careful attention to this point, which allows us to clarify the key distinctions of our work:
>
> 1) Our statement in the original submission aimed to highlight differences in *communication architectures* between the existing work and CALL (this paper), rather than  latent representation. While [R1,R2] indeed operate in latent space (as is the case in CALL), they employ centralized or static communication schemes where information is shared among *all agents*. In contrast, CALL introduces a dynamic, prediction-driven communication mechanism that selectively activates information sharing for distributed (ego-centric) learning.
>
> 2) The comparison is conducted among world model based approach (as in our statement "recent works on WM-based RL...") such that the use of latent representations is  a key technique across recent world model-based methods, including the cited works and CALL. However, the communication overhead in the  approaches in the cited works  scales significantly with network size due to their communication architecture. For instance, in our experimental evaluation with 230 agents (Appendix E), gathering all agents' latent information for centralized training requires bandwidth of nearly 50 times more than CALL's selective communication approach (as demonstrated in Figure 5). This substantive reduction in communication overhead, while maintaining performance, is a
> contribution of our work.
>
> We have revised the manuscript to better articulate this distinction between latent representation and communication architecture, ensuring our contribution is more precisely positioned within the literature.
>
> - [R1] Pan et al., "Iso-dream: Isolating and leveraging noncontrollable visual dynamics in world models.", NeurIPS 2022
>
> - [R2] Liu et al., "Efficient multi-agent reinforcement learning by planning", ICLR 2024
>
>
> **A2. Scalability and Adaptability** We thank the reviewer for your careful consideration of scalability and adaptability concerns, which gives an opportunity to better articulate the distributed nature of our approach. In fact, Scalability and Adaptability are two distinct advantages of CALL (this work) distinguishing  this proposed algorithm from existing centralized or static mechanisms.
>
> We clarify CALL's fundamental design principles as follows, which directly address the concerns.
>
> We first clarify that CALL is inherently a distributed learning framework where each agent trains *independently with its own local world model*, which means there is *no need* to determine the total number of agents beforehand. In particular, as is  standard [R3,R4], during the training stage, agents obtain a world model by leveraging privileged information such as BEV. As stated in line 183 and line 207, during testing, each agent leverages their own world model and the shared information for planning, which again does not need any predefined parameters on the numbers of agents.
>
> Second, we clarify that different from [R1,R2], the changes in the numbers of agents will NOT result in scalability issue. Specifically, CALL is an ego-centric distributed learning method based on a learned world model of its local environment which depends on only the agents in the proximity, which would not have scalability issue.  In particular, in our implementation, the shared information will all be fused into a unified BEV (with the same mask) first for planning and hence the agent doesn't need any change to its model architecture to cope with the increase of agents in the environment.
>
> Third, regarding computational complexity, our distributed approach actually offers superior scalability compared to centralized methods, as expected. Since each agent trains and operates independently, the computational load is naturally distributed across agents, which   is fundamentally different from centralized approaches [R1, R2]. Moreover, each agent only processes in its proximity, making the per-agent computation efficient.
>
> - [R3] Li et al. "Think2drive: Efficient reinforcement learning by thinking in latent world model for quasi-realistic autonomous driving (in carla-v2).", 2024
>
> - [R4] Chen et al., "Learning to drive from a world on rails." ICCV. 2021.

---

> ### Author Response · Authors · 2024-11-19
> **Reply to Reviewer nMor (2/2)**
>
> **A3. Latent Intention**
>
> The definition of intention in autonomous driving refers to the planned trajectory or waypoints that the vehicle aims to follow in the near future [R4,R5]. In our framework, latent intentions are derived from waypoint planning, as  mentioned in line 410. Specifically, we leverage CARLA's API for waypoint planning, following established practices in autonomous driving literature. While equations 11-13 appear to omit explicit references to intentions and information sharing, these elements are actually encoded within the latent state $x$ for notational simplicity. We will revise the manuscript to explicitly articulate this relationship and provide a more comprehensive description of how latent intentions are learned and utilized."
>
>
> - [R5] Casas, et al., "Intentnet: Learning to predict intention from raw sensor data." Conference on Robot Learning. PMLR, 2018.
>
> - [R6] Claussmann et al. "A review of motion planning for highway autonomous driving." IEEE Transactions on Intelligent Transportation Systems (2019)
>
>
>
>
>  **A4. Comparative Experiments**
>
> We appreciate the reviewer's suggestion regarding additional comparisons. Our choice of baselines was guided by several important considerations:
>
> First, we focused on world model-based approaches specifically designed for autonomous driving tasks, given the unique challenges of the high-dimensional CARLA environment. Many conventional RL approaches struggle with the curse of dimensionality in such settings without substantial modifications (as noted in Line 42). We used DreamerV3 as our primary baseline (denoted as 'Local Obs.' in Figure 3(a)) as it represents the state-of-the-art in world model-based RL. Additionally, we included a variant without waypoint sharing (LSI) to isolate the impact of our communication mechanism.
>
> While the works by [R1, R2] are world model based method, they were developed for fundamentally different environments, i.e., the DeepMind Control Suite and SMAC benchmark respectively. Adapting these methods to CARLA's autonomous driving setting would require significant architectural modifications that could compromise their original design principles. For instance, both [R1,R2] do not have dedicated module for intention process, which is critical for autonomous driving to understand the potential actions of other agents in the environment (and to mitigate the non-stationarity).
>
> To ensure fair comparison, we believe it's more appropriate to compare against methods specifically designed for similar autonomous driving scenarios, and in this case, Think2drve (Dreamerv3 based) approach is the SOTA on solving planning in CARLA benchmark. To our knowledge, CALL represents the first multi-agent world model-based approach specifically designed for autonomous driving tasks. We will revise the manuscript to better articulate these benchmark selection criteria.
>
>
> **A5. Experimental Verification of Accumulative Error in Multi-step Prediction**
>
> We thank the reviewer for this constructive suggestion regarding multi-step prediction analysis. While our current results focus on demonstrating the impact of information sharing on prediction error, we indeed provide the multi-step prediction visualtion in Figure 3, 13, 14,15,16 in Appendix E.4. In particular, we show the relationship between prediction accuracy and information (e.g., local information only) in Figures 13. We further provide the quantification results in our revision (\textbf{Appendix E.4})
>
>
>  **A6. Prediction Accuracy-Driven Information Sharing**
>
> We appreciate the opportunity to clarify our adaptive information sharing mechanism. As detailed in Section 3.3, CALL implements a dynamic communication scheme based on prediction accuracy:
>
> - Step 1: Each agent continuously monitors its prediction performance by comparing predicted latent states and intentions against actual observations over the past $K$ time-steps.
>
> - Step 2: When prediction errors exceed a threshold $c$, the agent automatically increase its communication range by 5 meters and initiates selective information exchange with relevant neighboring agents. Otherwise, the agent will remain its current communication range for information exchange.
>
> This adaptive approach ensures that communication remains both minimal and targeted, and agents only share information when and with whom it's most crucial for accurate prediction. We enhanced Section 3.3 with additional details and concrete examples to better illustrate this mechanism in our revision.
>
> ---
>
>
> We hope that we clarified these very helpful comments. We would be glad to engage in discussion if there are any other questions or parts that need further clarifications.

---

> > ### Comment · Reviewer_nMor · 2024-11-25
> >
> > In terms of comparative experiments, in fact, some papers are also required to supplement experiments on multi-agent navigation in non autonomous driving fields, which is even more unreasonable. In my opinion, experiments that only supplement relevant methods are reasonable and necessary. Although many benchmark algorithms are not developed in specific fields, they have undergone extensive testing in multi-agent environments, and appropriate and representative benchmark algorithms should be selected for comparison.

---

> > ### Comment · Reviewer_nMor · 2024-11-25
> > **Prediction Accuracy-Driven Information Sharing**
> >
> > You claim that this method can increase its communication range by 5 meters when prediction errors exceed a threshold , then can you validate the mechanism of your method through curve visualization?

---

> > ### Comment · Reviewer_nMor · 2024-11-25
> >
> > The accumulative prediction error is best reflected intuitively through the curves.

---

> ### Author Response · Authors · 2024-11-25
> **Reply to Reviewer nMor**
>
> We sincerely appreciate Reviewer nMor's original feedback and the time you've invested in reviewing our manuscript.
>
> We hope that our responses have adequately addressed your comments and concerns.
>
> As the rebuttal period is drawing to a close, we would like to check if there are any additional questions or points that would benefit from further clarification.
>
> We would be very happy to continue the discussion and provide any necessary additional information.

---

> ### Comment · Reviewer_nMor · 2024-11-25
>
> My question includes four aspects: observable scalability, testing adaptability, training transferability, and total quantity scalability. We know that there are three classic paradigms: centralized, decentralized, and shared. The centralized approach will inevitably have the problem of curse of dimensionality, while the decentralized approach of **establishing local models for each agent will inevitably lead to an increase in the number of model parameters as the number of agents increases.** Although shared manner can solve the above problems, it will face the disadvantage of homogenization and poor representation ability, which can easily lead to local optima. Moreover, in our opinions, the methods presented in the article did not bring any new insights, namely a very simplistic approach.

---

> > ### Comment · Reviewer_nMor · 2024-11-25
> >
> > And how does it perform compared to other shared and decentralized methods?

---

> ### Comment · Reviewer_nMor · 2024-11-25
>
> I want to know where intention w comes from? How to generate and update them?

---

> ### Author Response · Authors · 2024-11-27
> **Reply to Reviewer nMor's Follow-up Questions (1/2)**
>
> We thank the reviewer for engaging in the discussion. Below are our detailed responses to each concern:
>
> **A1. (Ego-centric MARL)**
>
> We first clarify that our work focus on **ego-centric MARL** (title and line 38) [R1,R2], where each agent chooses  actions  to maximize her *own interest*.  Grounded by our key results in Theorem 1 and Proposition 1,  our adaptive communication mechanism guide agents receive critical information when needed in order to minimize the sub-optimality gap. Moreover, theoretically, shared parameters in neural networks used for MARL can converge to globally optimal solutions [R3,R4] under certain conditions and such schemes has been widely used in practice . CALL consider each agent has their local world model, which processes agent-specific observations and intentions, ensuring heterogeneous behavior despite shared parameters.
>
> **(New Insights from CALL)** The CALL provide both **theoretical understanding** of MARL and a **practical solution**for autonomous driving applications, in particular,
>
>
> - **World Model based MARL**. We propose CALL to demonstrate the significant benefits of integrating world models into MARL for autonomous driving applications. As stated in line 42-53, the generalization capability of the WM is especially desired for applications like autonomous driving, where the ego vehicle needs to interact within the intricate dynamics. Meanwhile, as stated in line 57-71, the usage of WM can also overcome the notorious challenges of dimensionality and communication overhead induced in previous works.
>
> - **Prediction-guided lightweight information sharing**. This adaptive approach optimizes the critical trade-off between communication overhead and performance, ensuring robust coordination while minimizing bandwidth usage. Furthermore, CALL implements efficient communication by directly sharing critical information such as intentions and latent states, eliminating the need for complex communication modules. This lightweight approach maintains low computational overhead, making it particularly valuable for resource-constrained autonomous systems.
>
> - **Challenging evaluation domains**. We validate CALL in CARLA, a high-fidelity autonomous driving simulator, demonstrating its effectiveness under realistic conditions including complex traffic dynamics, uncertain agent behaviors, and real-world driving rules and constraints. Our experimental results show significant improvements over existing methods while maintaining practical deployability. The validation in CARLA is particularly meaningful as it subjects our method to strict safety and efficiency requirements that mirror real-world autonomous driving challenges. Through these comprehensive evaluations, we demonstrate that CALL not only advances the theoretical understanding of MARL but also provides a practical solution for autonomous driving applications.
>
> - [R1] Asuman et al. Independent learning in stochastic games. arXiv preprint arXiv:2111.11743, 2021.
>
> - [R2] Zhang et al, "Coordinating multi-agent reinforcement learning with limited communication." AAMAS, 2013.
>
>
> - [R3] Agazzi et al. "Global Optimality of Softmax Policy Gradient With Single Hidden Layer Neural Networks in the Mean-Field Regime." ICLR 2021
>
> - [R3] Hu et al. "Scalable multi-agent reinforcement learning for dynamic coordinated multipoint clustering." ToC 2022
>
> **A2. Experiments.**
>
>  (Baseline Choice) we clarify that our work specifically targets **autonomous driving**, where the complexity of road rules, physical constraints, and safety requirements create unique challenges distinct from general multi-agent environments. We deliberately chose Think2Drive as our primary baseline as it represents the current state-of-the-art in autonomous driving planning and has been extensively validated in the CARLA platform. The CARLA simulator is the standard testing environment in autonomous driving research, offering realistic physics, complex traffic scenarios, and standardized metrics. While there are many multi-agent algorithms in other domains, they typically lack the specialized components needed for handling structured road environments, traffic rules, and vehicle dynamics. This makes direct comparisons less meaningful for our specific use case.
>
> Meanwhile, we would be very interested in learning about specific benchmarks or model-based method that the reviewer has in mind that have demonstrated successful validation in CARLA while maintaining comparable functionality to our method.
>
>
> **A3. Learning curve.**
>
> We provide the curve visualization on how does the communication range change while the prediction errors exceed a threshold in our revision in Appendix I. We provide the comparison among three communication settings: CALL, Local observation only and full observation, and two scales: 150 agents and 250 agents.

---

> > ### Author Response · Authors · 2024-11-27
> > **Reply to Reviewer nMor's Follow-up Questions (2/2)**
> >
> > **A4. Where intention comes from?**
> >
> > (Intention) The intentions (waypoints) are generated through CARLA's built-in waypoint system, which provides a structured representation of the road network.  The initial intention extraction from current state and goal using an MLP encoder.
> >
> > Continuous updates based on:
> >
> > - Current observation (location)
> >
> > - Local planning objectives (e.g., destination)
> >
> > The intention is updated every few timestep to reflect changing dynamics and goals. The specific code are open sourced and available at [R3].
> >
> > [R3] {https://carla.readthedocs.io/en/latest/core_map/#waypoints}
> >
> > **A5. Comparison with other methods**
> >
> > Thank you for the question regarding comparative evaluation. Our work specifically on applying Ego-centric MARL to autonomous driving, a domain with *well-defined technical requirements and established testing environments*.
> >
> > Our comparative analysis operates on two levels.
> >
> > - First, as shown in Table 2, we examine existing shared and decentralized methods in the literature. While these methods typically focus on general control or gaming tasks, adapting them to autonomous driving scenarios would require *substantial modifications* to handle high-dimensional visual inputs and address non-stationary behavior inherent in multi-agent driving scenarios. The key distinction is that our method incorporates essential components specifically designed for autonomous driving, which notably the *latent world model* for processing high-dimensional sensory inputs and the *intention communication* for handling non-stationary agent behaviors.
> >
> > - Second, we conduct extensive empirical evaluation within the CARLA simulator, which provides an established testing environment with well-defined technical requirements for autonomous driving. We benchmark against Think2Drive (based on Dreamer v3), a state-of-the-art autonomous driving algorithm, across three different information sharing mechanisms: local observation only, shared state, and shared state with intention.
> >
> > We believe our domain-specific evaluation addresses the core technical challenges of autonomous driving while validating our method's effectiveness through comprehensive comparisons against state-of-the-art baselines in this domain and different information sharing mechanisms in standardized autonomous driving environments.
> >
> > **A6. Accumulative prediction error.**
> >
> > We add the prediction error curve in *Appendix H* in the revision, where we plot both the accumulation error and single-step prediction error for three communication settings: CALL, Local observation only and full observation.
> >
> > We hope our responses have sufficiently addressed the issues raised. If there are any remaining concerns requiring further clarification, we would be happy to provide additional explanations or make further adjustments as needed. Thank you again for your time and effort.

---

> ### Author Response · Authors · 2024-12-01
> **A Gentle Reminder**
>
> Dear Reviewer,
>
> As the rebuttal period is nearing its conclusion, we would like to kindly follow up to see if there are any additional questions or areas that need further clarification. Thank you once again for your time and valuable feedback.
>
> Authors

---

### Official Review · Reviewer_DLBj · 2024-11-04

**Soundness:** 3
**Presentation:** 3
**Contribution:** 3
**Rating:** 6
**Confidence:** 3

**Summary:**

This paper presents CALL (Communicative World Model), a framework for addressing partial observability and non-stationarity challenges in multi-agent reinforcement learning (MARL) in high-dimensional environments. The key innovation lies in synergizing world models' generalization capabilities with lightweight information sharing. Each agent encodes its state and intentions into low-dimensional latent representations and selectively shares them with other agents based on prediction accuracy. The approach is theoretically analyzed and validated on the CARLA autonomous driving platform.

**Strengths:**

The paper makes several significant contributions to multi-agent reinforcement learning. The core innovation of synergizing world models with lightweight communication is both novel and practical, addressing two fundamental challenges in MARL: partial observability and non-stationarity. The technical approach is well-grounded in theory, with Theorem 1 providing a detailed analysis of prediction error structure and Proposition 1 establishing bounds on sub-optimality gaps. The implementation is particularly impressive, demonstrating substantial performance improvements in the CARLA autonomous driving platform - reducing communication bandwidth from 5MB to 0.11MB while increasing success rate from 52% to 87% (Table 5). The ablation studies thoroughly validate each component's contribution, and the method shows good generalization to unseen environments. The practical relevance of the work is clear, with the framework successfully handling realistic scenarios involving up to 250 vehicles while maintaining reasonable computational requirements.

**Weaknesses:**

1. The experimental validation is confined to autonomous driving scenarios, raising questions about generalizability
2. Lack of comparisons with state-of-the-art MARL methods is a significant omission. Specifically: No comparison with recent methods like QMIX, MADDPG, or other communication-based approaches; No evaluation on standard MARL benchmarks; Missing comparison with other world model-based methods
3. The ablation studies, while thorough for the proposed components, don't explore alternative design choices
4. The world model training process is not fully described in the main text: 1) Missing details about the training curriculum; 2) Unclear how the model handles different types of sensory inputs; 3) Limited discussion of failure cases and their analysis
5. The prediction-accuracy-driven communication mechanism needs more elaboration: 1) The threshold selection process is not well-explained; 2) The adaptation mechanism for the communication range isn't fully specified; 3) Missing analysis of communication overhead in different scenarios
6. The bounds in Theorem 1 might be loose, and there's no discussion of their tightness: 1) No lower bounds are provided for comparison; 2) The analysis assumes finite action spaces, limiting generality; 3) The impact of approximation errors isn't fully analyzed
7. Missing formal analysis of communication complexity: 1) No theoretical guarantees on the optimality of the information sharing strategy; 2) Limited analysis of the trade-off between communication cost and performance

**Questions:**

1. How is the prediction accuracy threshold c determined? Is it static or dynamically adjusted?
2. How does the system handle communication delays and packet losses in practice?
3. What's the maximum number of agents the communication protocol can efficiently handle?
4. Are the bounds in Theorem 1 tight? Can a lower bound be provided?
5. How does the finite action space assumption affect application to continuous action spaces?
6. Can the theoretical analysis be extended to handle heterogeneous agents?
7. Why weren't standard MARL benchmarks included in the evaluation?
8. Can performance comparisons with methods like MADDPG or QMIX be provided?
9. Is 250 vehicles the scalability limit? What's the performance in larger-scale scenarios?
10. What are the computational requirements for training the world model?

---

> ### Author Response · Authors · 2024-11-19
> **Reply to Reviewer DLBj (1/3)**
>
> We thank the reviewer BEZm for your careful reading and constructive feedback. We address the concerns raised as follows.
>
> ### Weakness
>  **A1. Testing Environment.** Our work mainly focus on the planning tasks of autonomous driving in the multi-agent environment. In this regard, our choice of CARLA as the primary testing environment is justified by its wide adoption in the research community and real-world relevance. CARLA presents substantially more challenging scenarios compared to traditional MARL benchmarks, featuring realistic vehicle dynamics, complex multi-agent interactions, and strict traffic safety protocols. Most importantly, CARLA requires longer-horizon planning (3-5 seconds ahead) compared to shorter planning horizons in other benchmarks. While CALL's core principles are indeed applicable to other multi-agent scenarios, the success in this challenging environment provides strong evidence for its effectiveness in other simpler settings.
>
>  **A2. Baselines** We appreciate the reviewer’s suggestion regarding additional comparisons. Our choice of baselines was guided by several important considerations:
>
> First, we focused on world model-based approaches specifically designed for autonomous driving tasks, given the unique challenges of the high-dimensional CARLA environment. Many conventional RL approaches struggle with the curse of dimensionality in such settings without substantial modifications (as noted in Line 42). We choose the SOTA work [R1] on autonomous driving planning, which is based on DreamerV3,  as our primary baseline (denoted as 'Local Obs.' in Figure 3(a)). Additionally, we included a variant without waypoint sharing (LSI) to isolate the impact of our communication mechanism.
>
> The works such as QMIX, MADDPG, or other communication-based approaches either not using world model (hence not being able to effectively deal with high-dimensional inputs in CARLA), lack of intention sharing (which is essential for planning) or have requirements on for sharing all information among agents (hence impractical for a large multi-agent systems as considered in our work).  While the works by [R2, R3] are world model based method, they were developed for fundamentally different environments, i.e., the DeepMind Control Suite and SMAC benchmark respectively. Adapting these methods to CARLA's autonomous driving setting would require significant architectural modifications that could compromise their original design principles. For instance, both [R1,R2] do not have dedicated module for intention process, which is critical for autonomous driving to understand the potential actions of other agents in the environment.
>
> To ensure fair comparison, we believe it's more appropriate to compare against methods specifically designed for similar autonomous driving scenarios, and in this case, Think2drive (Dreamerv3 based) approach is the SOTA on solving planning in CARLA benchmark. To our knowledge, CALL represents the first multi-agent world model-based approach specifically designed for autonomous driving tasks. In our revision, we further articulate these benchmark selection criteria (ref.Section 4).
>
>
> - [R1] Li et al. "Think2drive: Efficient reinforcement learning by thinking in latent world model for quasi-realistic autonomous driving (in carla-v2).", 2024
>
> - [R2] Pan et al., "Iso-dream: Isolating and leveraging noncontrollable visual dynamics in world models.", NeurIPS 2022
>
> - [R3] Liu et al., "Efficient multi-agent reinforcement learning by planning", ICLR 2024
>
>  **A3. Ablation studies.** Our ablation studies were specifically designed to validate our key technical contributions (comparing with baseline SOTA method) and theoretical insights. They systematically evaluate the impact of state sharing on partial observability and intention sharing on non-stationarity, directly addressing our main technical claims. The studies demonstrate the effectiveness of our lightweight information sharing approach and validate our theoretical predictions about prediction accuracy and communication efficiency.
>
>  **A4. World model training.** In Appendix E, we provide comprehensive information about the training process, including the curriculum design, handling of multi-modal sensory inputs (camera, LiDAR) through BEV representation, and network architecture specifications. We also include analysis of failure cases and their implications for real-world deployment.

---

> ### Author Response · Authors · 2024-11-19
> **Reply to Reviewer DLBj (2/3)**
>
> **A5. Communication mechanism.** The communication mechanism operates with a carefully selected threshold c, which we determined through validation. Our implementation dynamically adjusts communication range based on prediction accuracy. When k-step prediction error exceeds threshold c, the range incrementally expands to include the next nearest neighbor until either prediction improves or maximum range is reached. The range contracts when predictions remain accurate, ensuring minimal but sufficient communication. We show that this approach achieves significant efficiency, requiring only 0.106MB average bandwidth compared to 5.417MB for full observation. In our revision, we add detailed quantitative results in Appendix E demonstrating how different thresholds affect performance and communication overhead.
>
>
>  **A6. Bounds in Theorem 1.** The upper bounds in Theorem 1 are tight for the case with two distributions for approximation errors. While deriving lower bounds remains challenging due to the non-trivial nature of RNN generalization bounds, this doesn't impact the practical utility of our results. The finite action space assumption follows standard practice in theoretical MARL analysis, and our implementation handles continuous actions through discretization, as demonstrated successfully in DreamerV3. We have clarified the impact of approximation errors on performance in Section 3.2.
>
>
>  **A7. Communication Complexity** While deriving theoretically optimal communication strategies is highly non-trivial due to the complexity of multi-agent interactions, we provide comprehensive empirical analysis of the accuracy-performance trade-offs. Our results demonstrate that CALL only requires 50 times less bandwidth than centralized case while maintaining high performance. The revision includes detailed quantitative analysis in \textbf{Appendix G} showing how different accuracy thresholds affect both system performance and communication overhead in autonomous driving scenarios.
>
> ### Questions
>
>  **A1. Prediction accuracy threshold.** The threshold c is a static hyperparameter that we determined through empirical validation. In our revision, we include comprehensive quantitative results in Appendix F  demonstrating how different threshold values affect the trade-off between system performance and communication overhead. This analysis provides a clear guidance for parameter selection in practical deployments.
>
>  **A2. Communication delays** While our current implementation assumes reliable communication, CALL's framework naturally accommodates extensions for handling delays and packet losses. The world model's prediction capability allows agents to continue operating even with temporary communication disruptions by relying on their learned dynamics models. In future work, we plan to incorporate explicit mechanisms for handling delays through prediction-based compensation and implement robust protocols for packet loss recovery.
>
>
>  **A3. Maximum number of agents.** CALL's communication protocol is inherently scalable because it's ego-centric and distributed. Each agent's observation space remains constant regardless of the total number of agents in the system, as it only depends on the agent's local sensing range. When an agent encounters others within its observation range, it dynamically decides whether to request information sharing based on prediction uncertainty. This approach doesn't require changes to model architecture or observation dimensions as the system scales, making it theoretically capable of handling any number of agents within bandwidth constraints.
>
>  **A4. Bounds in Theorem 1.** The upper bound in Theorem 1 is tight as the quality can be achieved when the equality holds in Assumptions 1 (Action and policy bound), 2 (weight matrices bound) and $\mathbf{E}\_{\pi}[{D}\_{\operatorname{TV}}(P || \hat{P})] = \mathcal{E}\_P$ (line 274). As can be seen in Eqn (10), once those equalities hold, the resulting upper bound derived in Theorem 1 will be achieved.  While lower bounds can provide theoretical completeness, the primary focus in generalization error analysis literature, e.g., [R1,R2,R3,R4] is on establishing upper bounds, as these directly inform algorithm design and practical implementation.
>
> - [R1] Wu et al., "Statistical machine learning in model predictive control of nonlinear processes", Mathematics, 2021
>
> - [R2] Lim et al., "Noisy recurrent neural networks", NeurIPS 2021
>
>
> - [R3] Tu et al. "Understanding generalization in recurrent neural networks." ICLR, 2020.
>
> - [R4 ] Chen et al., "On generalization bounds of a family of recurrent neural networks.", 2019

---

> ### Author Response · Authors · 2024-11-19
> **Reply to Reviewer DLBj (3/3)**
>
> **A5. Finite action space.** In practice, continuous action spaces can be effectively handled through discretization, following the successful approach demonstrated in DreamerV3. Our implementation discretizes the steering and acceleration actions while maintaining sufficient granularity for smooth control. This approach has proven effective in our CARLA experiments, achieving precise vehicle control without compromising performance. The theoretical results extend naturally to the discretized setting, making the finite action space assumption practical rather than limiting.
>
>
>  **A6. Heterogeneous agents** Yes, our theoretical framework can be extended to heterogeneous agents with additional assumptions. The key requirement would be that all agents' latent representations can be projected into a common latent space, allowing for meaningful information sharing across different agent types. This extension would require additional theoretical machinery to handle the mapping between different latent spaces, but the core principles of our analysis remain applicable.
>
>
>  **A7. Standard MARL benchmark** Please refer to our response in Weakness A1.
>
>
>
>  **A8. Baselines** Traditional MARL methods like MADDPG and QMIX face fundamental limitations in the CARLA environment. MADDPG struggles with high-dimensional visual inputs and doesn't scale well to large numbers of agents. QMIX, while effective for value function factorization, lacks mechanisms for handling intention sharing and requires centralized training. Adapting these methods to CARLA would require significant modifications that would compromise their original design principles.
>
>  **A9. Scalability limit** The 250-vehicle scenario is not a fundamental limit but rather a practical demonstration point. CALL's ego-centric distributed learning approach offers superior scalability compared to centralized methods. Since each agent trains and operates independently, the computational load scales linearly with the number of agents, unlike centralized approaches where complexity often scales exponentially. Each agent only processes its local observations and selective communications, making per-agent computation efficient and bounded regardless of the total system size.
>
>  **A10. Computational requirements** Each agent requires approximately 12 hours of training on an A100 GPU, comparable to single-agent training requirements. This efficiency stems from our distributed approach, where each agent trains independently, and the computational load is naturally distributed. While the total number of parameters scales linearly with the number of agents, this is fundamentally different from centralized approaches where complexity often scales exponentially. The per-agent computation remains constant and bounded, making the approach practical for large-scale deployments.
>
>
>
> We hope that we clarified these very helpful comments. We would be glad to engage in discussion if there are any other questions or parts that need further clarifications.

---

### Official Review · Reviewer_j41G · 2024-11-04

**Soundness:** 3
**Presentation:** 3
**Contribution:** 3
**Rating:** 6
**Confidence:** 3

**Summary:**

The paper tries to tackle the issues of partial observability and non-stationarity in multi-agent reinforcement learning (MARL) tasks in complex environments by developing a decentralized and adaptive communication pipeline. Verbatim from the paper, the authors presented an approach to answer: “How to synergize world model’s generalization capability with light-weight information sharing for enhancing ego-centric MARL in high-dimensional, non-stationary environments?”

The authors work on top of the Dreamer V3 architecture to learn a world model (WM) for compressing the sensor information and for learning the transition dynamics.
For light-weight communication, the authors propose to use the compressed latent representations to predict the next transitions and also to share the other agent’s latent state and latent intentions (waypoints).

They call this framework CALL (Communicative World Model), where every agent shares its latent state and intention (encoded using a learned WM) and tries to improve their WM by adaptive and effective sharing of information. This adaptive sharing of relevant latent information helps minimizing communication overhead and improves decision-making.
The authors presented some theorems and propositions to manifest how WM can improve the prediction performance and reduce sub-optimality gap.

**Strengths:**

- The paper tried to explain how prediction error can control the sub-optimality gap and present an upper bound on both these entities.

- The paper did a good ablation study (both in terms of experiments and theoretically) to investigate the impact of insufficient information sharing, locally sufficient information sharing, latent information sharing and full observation sharing.

- The authors presented the numbers of how much the bandwidth can be improved using the latent representations for information sharing.

- The paper has experiments to show that WM generalization combined with information sharing is an effective	component to improve prediction in distributed RL in high-dimension environments.

**Weaknesses:**

- The definitions, formulations, introduction of notations can be made a bit more coherent.

- Uncertainty related to information sharing was not discussed and taken care of in the paper and was mentioned as one of the problems in the introduction section as well.

- The authors also mention privacy-preserving techniques can also be integrated during information sharing to prevent any sensitive information being leaked.

**Questions:**

- Fig 1: Where is a_{1, t} and a_{2, t} coming from? I assume we can also have the notation for the policy from which these actions are getting sampled from.

- Fig1: Can we have the symbol for encoder be replaced by the conventional trapezoid? The current symbol gives an impression of a combined enc-dec model.

- Line 314: Can we please define PPAD?

- Line 318: Is that extra closing bracket here T_{i,t}) a typo?

- Line 377: ….3D environment…..: missing space after environment.

- Algorithm 1: Line 384: How is the communication range updated? Is there any update formula for that?

- Nit: Line 1322: “Large” could be made to “large”?

- Can we call this similar/analogous to the unification of trajectory prediction and planning task where the next best action during planning is conditioned on the predicted trajectories of other agents; here, the latent intention (encoded waypoints) are provided as information that is used to predict the state dynamics?

---

> ### Author Response · Authors · 2024-11-19
> **Reply to Reviewer j41G**
>
> ### Weakness
> We appreciate reviewer j41G for these thoughtful comments that help improve our manuscript's clarity and completeness. We address your concerns in our revision and our clarification is outlined as follows.
>
> **A1. Notation.**  In our revision, we introduce a consolidated notation section that clearly defines all variables and their relationships for clarity in Appendix B, Table 3. Specifically, we use consistent notation across sections, particularly in the transition from world model formulation to information sharing mechanism, making the mathematical flow more intuitive and easier to follow.
>
>
> **A2. Uncertainty.** We thank the reviewer for highlighting this important aspect of uncertainty during communication. We first clarify that in the introduction (line 044), the uncertainty refers to the agent's prediction error on other agents state and action intention, and such uncertainty can be mitigated through information sharing among agents. To this end, our current framework handles uncertainty through prediction-accuracy driven communication (where agents share information when prediction uncertainty exceeds a threshold). To avoid confusion, we formally give the definition of uncertainty in the context of prediction error (Section 3.1)
>
>
>
> **A3. Privacy.** We agree that the privacy preservation could be a very important future direction. Our method's selective communication mechanism naturally provides a foundation for privacy preservation, as agents only share compact latent representations rather than raw sensory data. In the revision, we elaborate on how specific privacy-preserving techniques (such as differential privacy or secure multi-party computation) could be integrated into our framework without compromising its core functionality (in Section 5). We believe a full implementation of these techniques warrants dedicated future work to properly address the complex trade-offs between privacy, performance, and computational efficiency.
>
> ### Questions
>
> **A1. Fig 1 notation $a$.** The action is sampled by using agent's current policy (ref. line 179).
>
> **A2. Fig 1 encoder.** We thank the author for pointing it out and we updated it in our revision.
>
> **A3. PPAD.** The PPAD (Polynomial Parity Arguments on Directed graphs) is formally defined through the 'End of the Line' problem on directed graphs, where each node has at most one predecessor and successor. PPAD is believed to be harder than P but easier than NP-complete. We include the definition in our revison.
>
> **A4. line 318.** Yes, we correct the typo in our revision.
>
>
> **A5. line 377** Thank you for pointing it out and we corrected it in our revision.
>
>
> **A6. Algorithm 1** The communication range is dynamically adjusted based on the agent's prediction accuracy. When the k-step prediction error exceeds threshold $c$, the range incrementally expands to include the next nearest neighbor until either the prediction improves or maximum range is reached. The range contracts when predictions remain accurate, ensuring minimal but sufficient communication for reliable performance. In our implementation, the distance between vehicles are known such that the expansion of the communication range is realized by including the next nearest vehicle's for information inquiring.
>
>
>  **A7. line 1322**  Thank you for pointing it out and we corrected it in our revision.
>
>
> **A8. Analogy to trajectory prediction** Yes, the analogy does make sense. The condition is indeed on the information shared by other agents.
>
> We hope that we clarified these very helpful comments. We would be glad to engage in discussion if there are any other questions or parts that need further clarifications.

---

> ### Comment · Reviewer_j41G · 2024-11-27
>
> Thanking the authors for addressing the comments.
>
> I went through the comments and discussions from other reviewers as well and it seems that the authors have tried to carefully address their concerns as well.
>
> The authors were able to explain their views regarding some common concerns related to the choice of benchmarks, communication range and scalability of the multi-agent framework.
>
> It would be nice if the addressed comments can get mentioned in the final version of the paper.
> Appreciation for the authors was putting efforts in this direction of combining world models and adaptive communication to tackle the problems with MARL. I would like to maintain my score considering the limitations pointed out by some other reviewers.
> Best wishes! :)

---

### Official Review · Reviewer_BEZm · 2024-11-07

**Soundness:** 2
**Presentation:** 3
**Contribution:** 2
**Rating:** 5
**Confidence:** 3

**Summary:**

This paper proposes incorporating the world model into the multi-agent communication protocol to address the partial observability and non-stationarity issues in multi-agent reinforcement learning (MARL). In particular, this paper provides theoretical justification for how partial observability and non-stationarity can affect the world model performance. It derives the prediction-accuracy-driven information-sharing strategy for better communication. The paper shows the performance of its method in the autonomous driving setting.

**Strengths:**

- The paper is clear and well-written. It is very easy for me to follow.
- I like the theoretical analysis and how it can inform a better communication strategy.
- It provides the experimental results on how the method can help to solve the two issues in MARL.

**Weaknesses:**

- This paper needs a comparison with other baselines (see Table 2) in your experiment result. Why is your proposed method superior to others? It is insufficient to show that you have specific components that others don't. The empirical results are essential to show that the method is effective.

- The experiment setting is limited. How would this method perform in other experiment settings besides CARLA?

**Questions:**

I suggest having a deeper analysis of your information-sharing mechanism in your experiment. I am curious: if your information-sharing mechanism is replaced by another mechanism, how would the performance change?

---

> ### Author Response · Authors · 2024-11-19
> **Reply to Reviewer BEZm (1/2)**
>
> ### Weakness
> We thank reviewer BEZm for your careful reading and feedback. We would like to clarify our experiments design in details as follows.
>
> **A1. Baseline Comparison.** We appreciate the reviewer’s suggestion regarding additional comparisons. Our choice of baselines was guided by several important considerations:
>
> First, we focused on world model-based approaches specifically designed for autonomous driving tasks, given the unique challenges of the high-dimensional CARLA environment. Many conventional RL approaches struggle with the curse of dimensionality in such settings without substantial modifications (as noted in Line 42). We choose the SOTA work just published in 2024 [R1] on autonomous driving planning, which is based on DreamerV3,  as our primary baseline (denoted as 'Local Obs.' in Figure 3(a)). Additionally, we included a variant without waypoint sharing (LSI) for ablation studies of the impact of our communication mechanism. To the best of our knowledge, CALL (this study) is the first multi-agent world model-based approach specifically designed for autonomous driving tasks.
>
> The works in Table 1 either not use world model (hence not being able to effectively deal with high-dimensional inputs in CARLA), or are lack of intention sharing (which is essential for planning),or have requirements on for sharing all information among agents (hence impractical for a large multi-agent systems as considered in our work).  While the works by [R2, R3] are world model based methods, they were developed for fundamentally different environments, i.e., the DeepMind Control Suite and SMAC benchmark respectively. Adapting these methods to CARLA's autonomous driving setting would require significant architectural modifications that could compromise their original design principles. For instance, both [R1,R2] do not have dedicated module for intention process, which is critical for autonomous driving to understand the potential actions of other agents in the environment.
>
> To ensure fair comparison, we believe it's more appropriate to compare against methods specifically designed for similar autonomous driving scenarios, and in this case, Think2drve (Dreamerv3 based) approach is the SOTA on solving planning in CARLA benchmark. To our knowledge, CALL is the first multi-agent world model-based approach specifically designed for autonomous driving tasks. In our revision, we further articulate these benchmark selection criteria (ref. Section 4).
>
>
> - [R1] Li et al. "Think2drive: Efficient reinforcement learning by thinking in latent world model for quasi-realistic autonomous driving (in carla-v2).", 2024
>
> - [R2] Pan et al., "Iso-dream: Isolating and leveraging noncontrollable visual dynamics in world models.", NeurIPS 2022
>
> - [R3] Liu et al., "Efficient multi-agent reinforcement learning by planning", ICLR 2024
>
> **A2. Method Superiority.**  Our empirical results demonstrate CALL's effectiveness through several key metrics:
> - (1) **Lightweight Information Sharing**. CALL uses well-crafted information sharing, which directly contributes to the significantly reduced communication overhead comparing with centralized setting as in [R2,R3] (about 50 times less bandwidth in 230-agent scenarios, ref. Figure 5).
> - (2) **Improved Generalization**. CALL leverages the received information (i.e., latent state and latent intention) and the world model's generalization capability to achieve improved prediction accuracy when compared with the SOTA method Think2drive [R1] (ref. Figure 3 and Figure 13).  Furthermore, in Figures 2 and 11, we show that CALL achieve better planning performance (in terms of the average return) thanks to the improved prediction accuracy.
> - (3) **Scalability**. CALL is an ego-centric distributed learning algorithm, where each agent trains their own world model for planning. We show that such approach can achieve stable performance scaling from 150 to 230 vehicles while maintaining per-agent efficiency in Figures 3 and 13.

---

> ### Author Response · Authors · 2024-11-19
> **Reply to Reviewer BEZm (2/2)**
>
> **A3. Experiments beyond CARLA.** We thank the reviewer's suggestions on broader experimental settings and we would like to clarify our choice of CARLA as our primary testing environment.
>
> CARLA presents substantially more challenging scenarios compared to traditional multi-agent benchmarks like DeepMind Control Suite and SMAC, particularly due to its realistic vehicle dynamics and multi-agent interactions that follow traffic rules and safety protocols. Meanwhile, the planning in CARLA generally needs longer-horizon and prediction (3-5 seconds ahead) versus shorter planning horizons as in other benchmarks.
>
> While CALL's core principles of distributed learning, prediction-driven communication, and ego-centric world models, are indeed applicable to other multi-agent scenarios, we chose autonomous driving as our primary test case due to its compelling combination of real-world significance and rigorous requirements for safety, efficiency, and scalability. The successful demonstration of CALL in this challenging environment provides strong evidence for its potential effectiveness in other multi-agent settings.
>
>
>
> ### Questions
> **A1. (information-sharing mechanism)**  We thank the reviewer for your suggestions on emphasizing the impact of the information-sharing mechanism. In our experiments, we conducted comprehensive comparisons between different communication strategies to demonstrate the effectiveness of our prediction-accuracy driven approach:
>
> **Case I.** No Communication: Agents rely solely on their local observations, which corresponds to the case with communication range is zero (or the case of accuracy threshold is large enough)
>
> **Case II.** Full Communication: Agents share information with all neighbors within range, which corresponds to the case with large communication range (or the case of prediction accuracy threshold is zero)
>
> **Case III.** Our Proposed Method: Selective communication based on prediction accuracy (with a finite value of prediction accuracy threshold).
>
> As can be seen in Figure 2(c) and Figure 9(b), our proposed method can (1) achieve better prediction and planning comparing with sharing no information and also (2) achieve the same prediction performance more efficiently compared to the full information sharing case with far less bandwidth requirements.
>
>
> Furthermore, our detailed analysis reveals a critical trade-off between prediction accuracy requirements and communication efficiency. Setting higher prediction accuracy requirements (lower error threshold) improves learning performance through more precise state estimation, but increases communication overhead in a larger communication range. Conversely, lower accuracy requirements reduce communication burden but can significantly degrade performance due to incomplete state information, particularly in complex scenarios requiring precise coordination. Through extensive experiments, we identified an operating point that balances these competing factors, achieving robust performance while maintaining efficient communication. In our revision, we present the quantitative results in **Appendix G** demonstrate this trade-off, showing how different accuracy thresholds affect both system performance and communication overhead in autonomous driving scenarios.
>
>
> We hope that we clarified these very helpful comments. We would be glad to engage in discussion if there are any other questions or parts that need further clarifications.

---

> ### Author Response · Authors · 2024-12-01
> **Gentle Reminder**
>
> Dear Reviewer,
>
> As the rebuttal period is nearing its conclusion, we would like to kindly follow up to see if there are any additional questions or areas that need further clarification. Thank you once again for your time and valuable feedback.
>
> Authors

---

### Meta-Review · Area_Chair_tGJJ · 2024-12-20

**Metareview:**

This paper proposed an ego-centric MARL method for autonomous driving, CALL. In this method, agents share their latent states and intentions encoded using a learned world model. They then improve their world models by sharing information. This method addresses the important problem of partial observability of agents in multi-agent systems. While the method is interesting, all reviewers raised concerns about the lack of evaluation:

1) The experiments were only done on CARLA.
2) There is no comparison against SOTA MARL methods.

The AC thinks that 1) is fine for a paper that only focuses on autonomous driving. However, the explanation provided by the authors about why there are no other SOTA MARL baselines is not sufficiently convincing. Even though they are not originally designed for autonomous driving, it doesn't mean that comparison is not possible or meaningful. The paper can benefit from further evaluation.

**Additional Comments On Reviewer Discussion:**

Two reviewers provided additional justification for their final recommendations of reject during the reviewer discussion. The concern about evaluation remains after the rebuttal and the AC-reviewer discussion.

---

### Decision · Program_Chairs · 2025-01-22

Reject